# Critical role for Piccolo in synaptic vesicle retrieval

**Frauke Ackermann[1]\*, Kay Oliver Schink[2], Christine Bruns[1], Zsuzsanna Izsvák[3], F Kent Hamra[4], Christian Rosenmund[5], Craig Curtis Garner[1]\***

[1]German Center for Neurodegenerative Diseases e.V. (DZNE), Charité Medical University, Berlin, Germany; [2]Center for Cancer Biomedicine, University of Oslo, Norwegian Radium Hospital, Oslo, Norway; [3]Max-Delbrück Center for Molecular Medicine in the Helmholtz Society, Berlin, Germany; [4]Department of Obstetrics and Gynecology, University of Texas Southwestern, Dallas, United States; [5]NeuroCure Cluster of Excellence, Neuroscience Research Center, Charité Medical University, Berlin, Germany

**Abstract** Loss of function of the active zone protein Piccolo has recently been linked to a disease, Pontocerebellar Hypoplasia type 3, which causes brain atrophy. Here, we address how Piccolo inactivation in rat neurons adversely affects synaptic function and thus may contribute to neuronal loss. Our analysis shows that Piccolo is critical for the recycling and maintenance of synaptic vesicles. We find that boutons lacking Piccolo have deficits in the Rab5/EEA1 dependent formation of early endosomes and thus the recycling of SVs. Mechanistically, impaired Rab5 function was caused by reduced synaptic recruitment of Pra1, known to interact selectively with the zinc finger domains of Piccolo. Importantly, over-expression of GTPase deficient Rab5 or the Znf1 domain of Piccolo restores the size and recycling of SV pools. These data provide a molecular link between the active zone and endosome sorting at synapses providing hints to how Piccolo contributes to developmental and psychiatric disorders.

DOI: https://doi.org/10.7554/eLife.46629.001

**\*For correspondence:**
Frauke.Ackermann@dzne.de (FA);
craig.garner@dzne.de (CCG)

**Competing interests:** The authors declare that no competing interests exist.

## Introduction

Piccolo is a scaffolding protein of the cytomatrix assembled at the active zone (CAZ), a specialized region within the presynaptic terminal where SV fusion takes place (*Ackermann et al., 2015*; *Gundelfinger et al., 2015*). It is a multi-domain protein, which is exclusively localized to active zones where it forms a sandwich-like structure enclosing Bassoon (*Nishimune et al., 2016*).

At present, the function of Piccolo is not well understood. Initial genome-wide association studies have linked missense mutations in Piccolo to psychiatric and developmental disorders including major depressive, bipolar disorder (*Choi et al., 2011*; *Giniatullina et al., 2015*; *Minelli et al., 2012*; *Sullivan et al., 2009*; *Woudstra et al., 2013*) and Pontocerebellar Hypoplasia type 3 (PCH3) (*Ahmed et al., 2015*). How an active zone protein like Piccolo contributes to these diseases is still unclear.

Cellular, biochemical and molecular studies indicate a structural role for Piccolo in organizing the active zone (AZ) (*Ackermann et al., 2015*; *Gundelfinger et al., 2015*). Loss of Piccolo in the retina disrupts the assembly of photoreceptor cell ribbons (*Müller et al., 2019*; *Regus-Leidig et al., 2014*). However, its association with the SV priming factors RIM1 and Munc13-1 (*Südhof and Rizo, 2011*) implies an additional role in SV priming, though a direct role in regulating neurotransmitter release has not been demonstrated (*Mukherjee et al., 2010*). Furthermore, structure/function studies have shown that Piccolo directly interacts with a number of actin-binding proteins, including Profilin-2, Daam1, Abp1, and Trio, and is critical for the activity dependent assembly of presynaptic F-actin

(*Fenster et al., 2003*; *Leal-Ortiz et al., 2008*; *Terry-Lorenzo et al., 2016*; *Wagh et al., 2015*; *Waites et al., 2011*; *Wang et al., 1999*). Moreover, it was also found to interact with proteins known to regulate endosome membrane trafficking like Prenylated Rab acceptor protein 1 (Pra1) (*Fenster et al., 2000*), and the Arf GTPase-activating protein 1 (GIT1) (*Kim et al., 2003*). Their inter-play suggests a possible role for Piccolo in the activity dependent SV membrane recycling and trafficking.

So far, endosome membrane trafficking within presynaptic boutons is only poorly understood. Early studies have shown endosome structures at presynaptic boutons (*Heuser and Reese, 1973*) as well as the presence of small Rab GTPases, including Rab4, 5, 7, 10, 11, 35, 26 and 14 (*Pavlos and Jahn, 2011*). Members of the small Rab GTPase family are organizers of membrane trafficking tasks, as they function as molecular switches alternating between a GDP bound 'off-state' and a GTP bound 'on-state' (*Stenmark, 2009*). In their 'active state', Rab GTPases can recruit effector proteins such as EEA1 to specific membrane compartments (*Christoforidis et al., 1999*; *Spang, 2009*). Often these effector proteins also bind to phosphoinositides (PIs), which are also present at synapses (*Uytterhoeven et al., 2011*; *Wucherpfennig et al., 2003*). Together, active small Rab GTPases and specific PI species define the identity of a compartment (*Balla, 2013*; *Schink et al., 2016*; *Zerial and McBride, 2001*).

Perturbation of endosomal proteins as well as over-expression of constitutive active Rab5 (Rab5$^{Q79L}$) affects synapse function (*Hoopmann et al., 2010*; *Rizzoli and Betz, 2002*; *Wucherpfennig et al., 2003*). Thus endosome membrane trafficking appears to be a fundamental feature of presynaptic boutons, helping them maintain functional fidelity over time.

To better understand the synaptic function of Piccolo, we recently generated a line of rats in which the *Pclo* gene was disrupted by transposon mutagenesis (*Pclo$^{gt/gt}$*) (*Medrano et al., 2018*). Our analysis of hippocampal synapses reveals that knockout of Piccolo causes a dramatic decrease in the number of SVs per synaptic bouton in combination with a smaller total recycling pool (TRP) of SVs and defects in the efficient reformation of SVs. Mechanistically, we find that these phenotypes are caused by the reduced recruitment of the Piccolo binding partner Pra1 to presynaptic boutons, the activation of Rab5 as well as the formation of early endosomes. These data provide a molecular link between the active zone protein Piccolo, endosome trafficking and the functional maintenance of SV pools providing a possible molecular mechanism for how Piccolo loss could contribute to PCH3 and major depressive disorders among others.

## Results

### Generation and characterization of the Piccolo knockout rat

Transposon mutagenesis was used to generate rats with a disrupted *Pclo* gene (*Pclo$^{gt/gt}$*) (*Medrano et al., 2018*). As shown in *Figure 1A*, the transposon element was integrated into exon 3 of the *Pclo* genomic sequence, leading to a premature stop in the reading frame. Genotypes of offspring were determined by PCR from genomic DNA (*Figure 1B*). Western blot analysis from post-natal day 2 (P2) rat brains as well as from primary hippocampal neuron lysates confirmed the loss of Piccolo full-length protein (*Figure 1C and D*). Bands at 560 kD as well as between 70–450 kD were detected in lysates from *Pclo$^{wt/wt}$* and *Pclo$^{wt/gt}$* brains (*Figure 1C*), reflecting the expression of multiple Piccolo isoforms from the *Pclo* gene (*Fenster and Garner, 2002*). In brain lysates from *Pclo$^{gt/gt}$* ani-mals, the dominant bands at 560 and 450 kD were absent, indicating the loss of the predominant Pic-colo isoforms, though a few weaker bands at 300, 90 and 70 kD are still present (*Figure 1C*). Notably, these are not present in lysates from primary hippocampal neurons (*Figure 1D*). Consistently, Piccolo intensity at synapses (positive for VGlut1) was reduced by more than 70% in *Pclo$^{gt/gt}$* neurons (*Figure 1E*), confirming the loss of most Piccolo isoforms from synapses. Similarly, Piccolo immuno-reactivity was absent in hippocampal brain sections (*Figure 1F*).

As Piccolo is a core protein of the AZ, we examined the effect of loss of Piccolo on the expression levels of other synaptic proteins. In Western blot analysis of P2 rat brain lysates, most proteins tested, were not affected (*Figure 1G*). Intriguingly, Piccolo deficiency was associated with a slight decrease in the expression of Bassoon and the SV protein Synaptophysin (*Figure 1G*). The levels of the early endosome marker EEA1 were most severely affected, its levels were decreased by half in *Pclo$^{wt/gt}$* and *Pclo$^{gt/gt}$* lysates (*Figure 1G*). In contrast, the levels of the small Rab GTPase Rab5 were

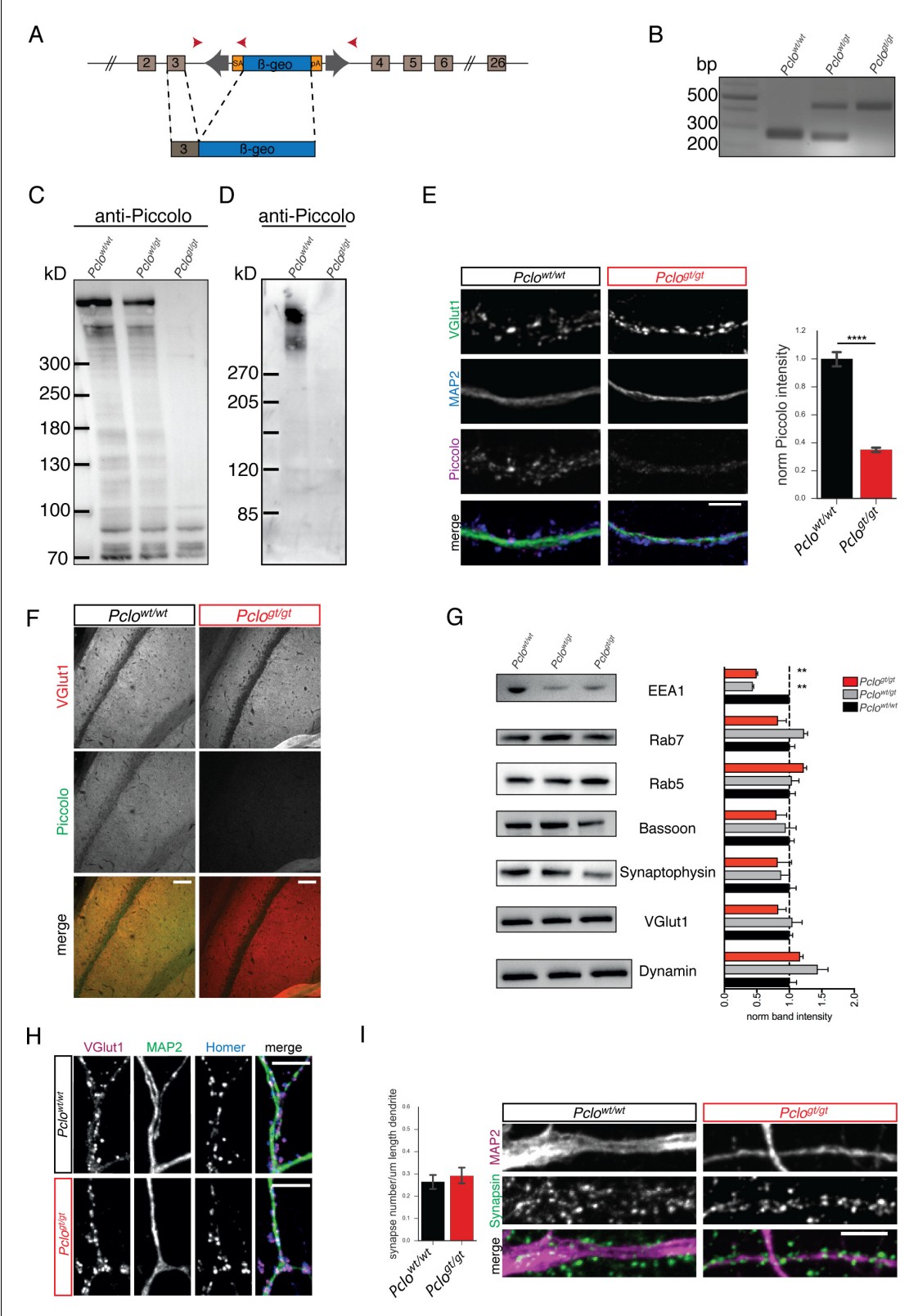

**Figure 1.** Transposon mutagenesis leads to the loss of Piccolo protein in rats. (A) Schematic of transposon insertion into exon 3 of *Pclo* genomic sequence creating a translational stop codon, truncating the generation of long Piccolo isoforms. (B) PCR verification of genotypes. PCR from *Pclo^wt/wt^*, *Pclo^wt/gt^* and *Pclo^gt/gt^* genomic DNA results in the amplification of 260 bp WT and/or 450 bp KO specific bands, respectively. (C and D) Western blot analysis of P0-P2 brain lysates (C) and lysates from DIV14 cultured hippocampal neurons (D). Bands, corresponding to Piccolo isoforms (70–560 kDa), *Figure 1 continued on next page*

*Figure 1 continued*

are detected in $Pclo^{wt/wt}$ and $Pclo^{wt/gt}$ brain lysates as well as $Pclo^{wt/wt}$ neuron lysates, but not in $Pclo^{gt/gt}$ lysates (n = 3 independent experiments). (E) Immunocytochemical staining of hippocampal neurons. Piccolo immuno-reactivity co-localizes with VGlut1 in $Pclo^{wt/wt}$ neurons, but not in $Pclo^{gt/gt}$ neurons. Right panel, quantification of Piccolo intensities at VGlut1 puncta ($Pclo^{wt/wt}$ = 1 ± 0.03, n = 586 synapses; $Pclo^{gt/gt}$ = 0.35 ± 0.01, n = 718 synapses; two independent experiments). (F) Immuno-histological staining of $Pclo^{wt/wt}$ and $Pclo^{gt/gt}$ brain sections. Representative images from the CA1 region of the hippocampus are shown. No Piccolo immuno-reactivity can be observed on $Pclo^{gt/gt}$ sections. (G) Western blot analysis of P0-P2 brain lysates. Expression levels of synaptic proteins are altered in $Pclo^{gt/gt}$ brain lysates (EEA1: $Pclo^{wt/wt}$ = 1 ± 0.03, $Pclo^{wt/gt}$ = 0.44 ± 0.01, $Pclo^{gt/gt}$ = 0.50 ± 0.02; two independent experiments; Rab7: $Pclo^{wt/wt}$ = 1 ± 0.08, $Pclo^{wt/gt}$ = 1.23 ± 0.05, $Pclo^{gt/gt}$ = 0.83 ± 0.13; four independent experiments; Rab5: $Pclo^{wt/wt}$ = 1 ± 0.09, $Pclo^{wt/gt}$ = 1.04 ± 0.11, $Pclo^{gt/gt}$ = 1.22 ± 0.04; three independent experiments; Bassoon: $Pclo^{wt/wt}$ = 1 ± 0.07, $Pclo^{wt/gt}$ = 0.95 ± 0.16, $Pclo^{gt/gt}$ = 0.80 ± 0.16; four independent experiments; Synaptophysin: $Pclo^{wt/wt}$ = 1 ± 0.11, $Pclo^{wt/gt}$ = 0.86 ± 0.13, $Pclo^{gt/gt}$ = 0.82 ± 0.21; three independent experiments; VGlut1: $Pclo^{wt/wt}$ = 1 ± 0.05, $Pclo^{wt/gt}$ = 1.05 ± 0.15, $Pclo^{gt/gt}$ = 0.83 ± 0.12; three independent experiments; Dynamin: $Pclo^{wt/wt}$ = 1 ± 0.11, $Pclo^{wt/gt}$ = 1.44 ± 0.16, $Pclo^{gt/gt}$ = 1.17 ± 0.05; three independent experiments). (H) Immunocytochemical staining of hippocampal neurons with antibodies against Homer, VGlut1 and MAP2 reveal the presence of VGlut1/Homer positive synapses along MAP2 positive dendrites in $Pclo^{wt/wt}$ and $Pclo^{gt/gt}$ neurons. (I) Images, of hippocampal neurons immunostained with antibodies against Synapsin and MAP2. The number of Synapsin positive puncta along primary dendrites (MAP2) does not differ between $Pclo^{wt/wt}$ and $Pclo^{gt/gt}$ neurons (left panel, Pclo^{wt/wt} = 0.27 ± 0.02, n = 27 primary dendrites; Pclo^{gt/gt} = 0.29 ± 0.02, n = 22 primary dendrites; two independent experiments). Scale bar in E, H and I 10 μm, scale bar in F 50 μm. Error bars in bar graph represent 95% confidence intervals. Numbers given represent mean ± SEM, Students T-test. **** denotes p<0.0001.

DOI: https://doi.org/10.7554/eLife.46629.002

The following source data and figure supplement are available for figure 1:

**Source data 1.** This spreadsheet contains the normalized values used to generate the bar plots shown in *Figure 1E,G and I*.
DOI: https://doi.org/10.7554/eLife.46629.004

**Figure supplement 1.** Levels of synaptic vesicle proteins are reduced at $Pclo^{gt/gt}$ synapses.
DOI: https://doi.org/10.7554/eLife.46629.003

slightly increased in $Pclo^{gt/gt}$ brain lysates, whereas the levels of Dynamin were increased in both $Pclo^{wt/gt}$ and $Pclo^{gt/gt}$ brain lysates (*Figure 1G*). However, these observed changes did not affect synapse formation, as VGlut1 puncta opposing Homer puncta were present in $Pclo^{wt/wt}$ and $Pclo^{gt/gt}$ neurons (*Figure 1H*) and the number of Synapsin puncta per unit length of primary dendrite was not significantly different between $Pclo^{wt/wt}$ and $Pclo^{gt/gt}$ neurons (*Figure 1I*).

Even though the number of synapses was not changed, the levels of several SV proteins were decreased in $Pclo^{gt/gt}$ neurons (*Figure 1—figure supplement 1 A, B, C, D*). These data indicate that Piccolo loss of function may lead to the loss of SVs from the presynaptic terminal. A conclusion further supported by the fact that the levels of the cytosolic SV-associated protein Synapsin were not affected by the loss of Piccolo (*Figure 1—figure supplement 1 A and E*).

## Ultrastructural analysis of $Pclo^{gt/gt}$ synapses

To more directly assess a role of Piccolo in synapse assembly, we analyzed the ultrastructure of synapses by electron microscopy (EM) using high pressure freezing and freeze substitution. EM micrographs from $Pclo^{wt/wt}$ and $Pclo^{gt/gt}$ neurons revealed the presence of synaptic junctions, with prominent postsynaptic densities (PSDs) formed onto axonal varicosities of $Pclo^{gt/gt}$ neurons, with similar dimensions (length) to $Pclo^{wt/wt}$ synapses (*Figure 2A,B,E*). At many $Pclo^{gt/gt}$ synapses, SVs (<50 nm diameter) were detected (*Figure 2B*), however the number of SVs/bouton was significantly reduced (*Figure 2A,B,C*), which is in line with the earlier observed reduced levels of Synaptophysin, Synaptotagmin and VGlut1 (*Figure 1—figure supplement 1*). Even though the total number of SVs was changed, the number of docked SVs at AZs was not altered in $Pclo^{gt/gt}$ synaptic terminals (*Figure 2D,H,I*). Intriguingly, lower SV density in $Pclo^{gt/gt}$ boutons was accompanied by a high number of endosome-like structures with diameters larger than 60 nm (*Figure 2A,B,F,G*). These data suggest that Piccolo is required for the maintenance and/or recycling of SVs.

## Pools of recycling SVs are reduced but synaptic release properties are not altered in boutons lacking Piccolo

Given Piccolo's restricted and tight association with presynaptic AZs (*Cases-Langhoff et al., 1996*), we also explored whether loss of Piccolo affects synaptic transmission. We performed whole cell patch clamp recordings from autaptic hippocampal neurons and observed a slight, but non-significant, reduction in the amplitudes of excitatory postsynaptic currents (EPSC) in $Pclo^{gt/gt}$ compared to

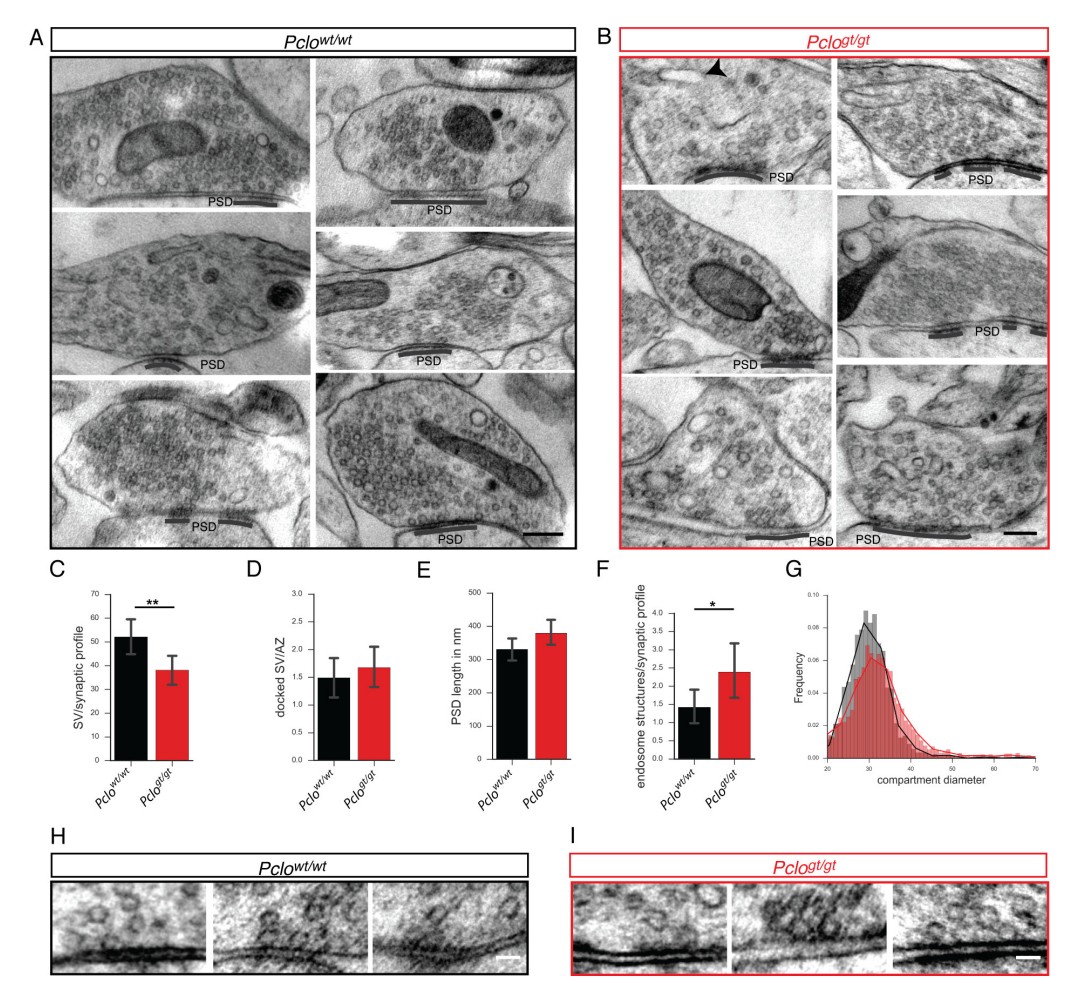

**Figure 2.** *Pclo^{gt/gt}* synapses display morphological changes at the ultrastructural level. (**A and B**) Example electron micrographs of *Pclo^{wt/wt}* (**A**) and *Pclo^{gt/gt}* (**B**) synapses. (**C–G**) Quantification of electron micrographs. (**C**) Synaptic vesicle density is decreased in *Pclo^{gt/gt}* terminals (*Pclo^{wt/wt}* = 51.73 ± 4.05, n = 52 synapses; *Pclo^{gt/gt}* = 38.08 ± 3.19, n = 59 synapses). (**D**) The number of docked vesicles per active zone is not altered (*Pclo^{wt/wt}* = 1.49 ± 0.18, n = 59 AZs; *Pclo^{gt/gt}* = 1.67 ± 0.18, n = 40 AZs). (**E**) The total length of the PSDs is not altered (*Pclo^{wt/wt}* = 331 ± 17.09 nm, n = 72 PSDs; *Pclo^{gt/gt}* = 379.6 ± 18.22 nm, n = 69 PSDs). (**F**) The number of endosome structures is increased at *Pclo^{gt/gt}* synapses (*Pclo^{wt/wt}* = 1.42 ± 0.24, n = 52 synapses; *Pclo^{gt/gt}* = 2.39 ± 0.38, n = 57 synapses). (**G**) Histogram depicting the distribution of endosome compartment diameters. *Pclo^{gt/gt}* vesicular compartments show a shift towards larger diameters (*Pclo^{wt/wt}*: n = 2729 compartments measured; *Pclo^{gt/gt}*: n = 2387 compartments measured). (**H and I**) Example electron micrographs showing docked vesicles at *Pclo^{wt/wt}* (**H**) and *Pclo^{gt/gt}* AZs (**I**). Scale bar in A and B 200 nm, scale bar in D and E 50 nm. Error bars in bar graph represent 95% confidence intervals. Numbers given represent mean ± SEM, Students *t*-test. * denotes p<0.05, ** denotes p<0.01.

DOI: https://doi.org/10.7554/eLife.46629.005

The following source data is available for figure 2:

**Source data 1.** This spreadsheet contains the normalized values used to generate the bar plots shown in *Figure 2C,D,E,F and G*.

DOI: https://doi.org/10.7554/eLife.46629.006

*Pclo^{wt/wt}* neurons (*Figure 3A*). Also the readily releasable pool of vesicles (RRP), determined by the application of hypertonic sucrose (5 s, 500 mM), was not altered in *Pclo^{gt/gt}* neurons (*Figure 3D*). This is consistent with the unaltered number of docked vesicles observed in EM analysis (*Figure 2D*). Similarly, SV release probability (Pvr) and paired pulse ratio (PPR) (25 ms pulse interval) were not changed in *Pclo^{gt/gt}* neurons (*Figure 3B and C*). However, the steady state EPSC amplitude at the end of a 10 Hz/5 s train stimulation was reduced in *Pclo^{gt/gt}* autapses, indicating impaired maintenance of synaptic transmission during intense vesicle fusion (*Figure 3E*). Together, these data indicate that neurotransmitter release properties of SVs are not severely changed at AZs lacking Piccolo,

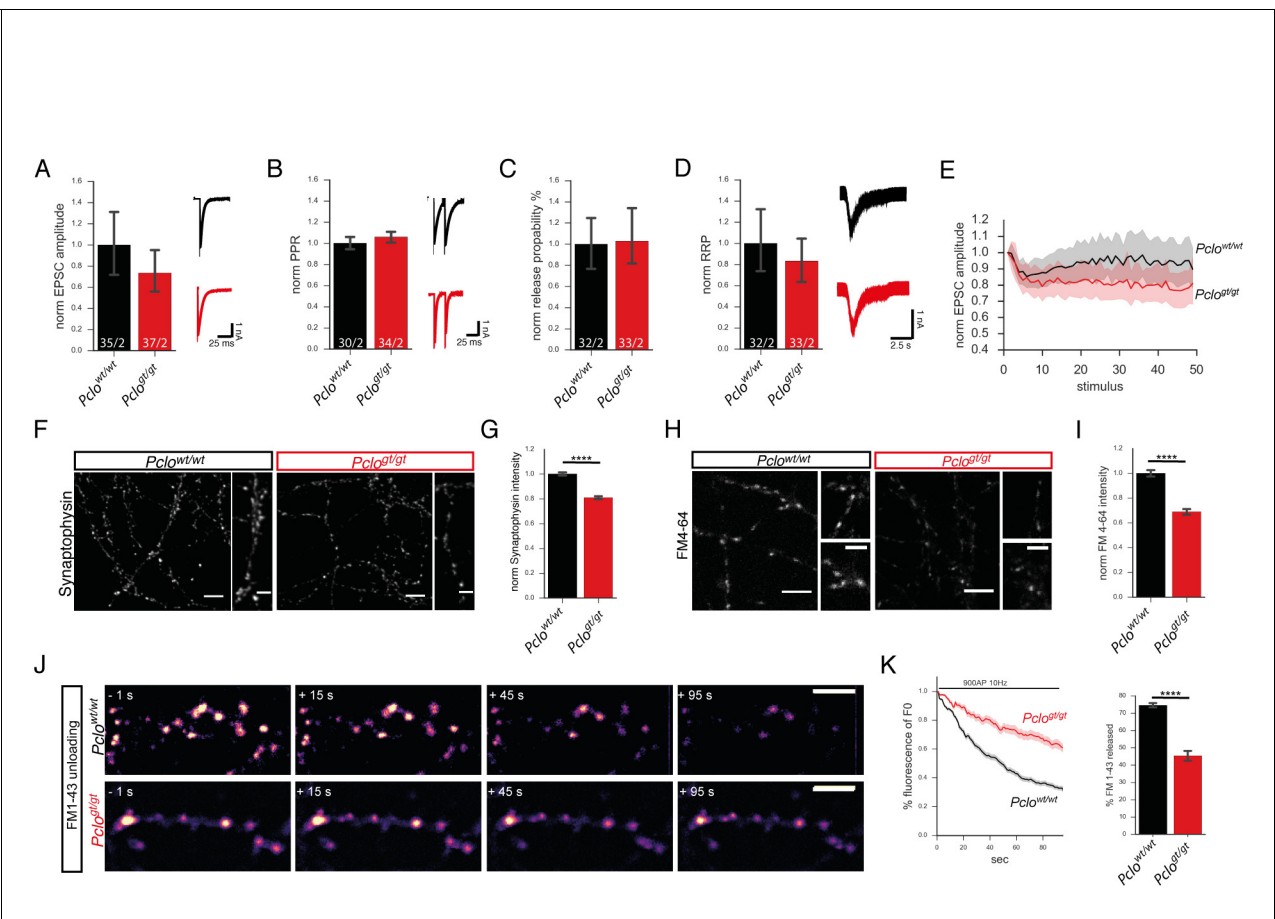

**Figure 3.** Synaptic vesicle release properties are not altered, but the total recycling pool of vesicles is reduced due to the loss of Piccolo. (A–E) Patch clamp recordings on autaptic primary hippocampal neurons. (A) Amplitudes of evoked postsynaptic currents (EPSCs) are slightly but not significantly reduced in $Pclo^{gt/gt}$ neurons ($Pclo^{wt/wt}$ = 1.0 ± 0.15, $Pclo^{gt/gt}$ = 0.73 ± 0.10; two independent experiments). (B) Paired pulse ratio is not altered in $Pclo^{gt/gt}$ neurons ($Pclo^{wt/wt}$ = 1.00 ± 0.03, $Pclo^{gt/gt}$ = 1.06 ± 0.03; two independent experiments). (C) Vesicle release probability is not changed upon Piccolo loss ($Pclo^{wt/wt}$ = 1 ± 0.13, $Pclo^{gt/gt}$ = 1.03 ± 0.13; two independent experiments). (D) The readily releasable pool of vesicles (RRP) in $Pclo^{gt/gt}$ neurons is not altered ($Pclo^{wt/wt}$ = 1 ± 0.15, $Pclo^{gt/gt}$ = 0.83 ± 0.11; two independent experiments). (E) Loss of Piccolo causes an increase in EPSC amplitude depression during 10 Hz train stimulation ($Pclo^{wt/wt}$ : 34 cells, $Pclo^{gt/gt}$ : 35 cells; two independent experiments). (F) Images of hippocampal neurons immuno-stained with Synaptophysin antibodies. (G) Quantitation of (F). Synaptophysin intensities/bouton are significantly decreased in $Pclo^{gt/gt}$ neurons ($Pclo^{wt/wt}$ = 1 ± 0.01, n = 3457 puncta; $Pclo^{gt/gt}$ = 0.81 ± 0.01, n = 2916 puncta; 13 independent experiments). (H) Images from FM4-64 dye uptake experiments. (I) Quantification of (H). FM4-64 dye uptake is significantly reduced in $Pclo^{gt/gt}$ neurons ($Pclo^{wt/wt}$ = 1 ± 0.01, n = 1026 puncta; $Pclo^{gt/gt}$ = 0.69 ± 0.01, n = 867 puncta; four independent experiments). (J) Selected images of synaptic boutons releasing loaded FM1-43 dye during a 900 AP 10 Hz stimulation. (K) Quantification of changes in FM1-43 dye intensities per bouton over time. Note, FM1-43 unloading rate is slower in $Pclo^{gt/gt}$ versus $Pclo^{wt/wt}$ neurons. In $Pclo^{wt/wt}$ neurons, about 70% of the initially loaded FM1-43 dye is released within 90 s of stimulation ($Pclo^{wt/wt}$ = 74.62 ± 0.6607, n = 244 synapses, three independent experiments), whereas in $Pclo^{gt/gt}$ neurons only about 45% is released ($Pclo^{gt/gt}$ = 45.47 ± 1.468, n = 155 synapses, three independent experiments). Scale bar in F, H and J 10 µm, scale bar in zoom in F and H 5 µm. Numbers in bar graphs (A–D) represent number of cells/number of cultures. Error bars in bar graph represent 95% confidence intervals. Numbers given represent mean ± SEM, Student's $t$ –test. **** denotes p<0.0001.

DOI: https://doi.org/10.7554/eLife.46629.007

The following source data is available for figure 3:

**Source data 1.** This spreadsheet contains the normalized values used to generate the bar plots shown in **Figure 3C,D,F,G,I and K**.
DOI: https://doi.org/10.7554/eLife.46629.008

**Source data 2.** This file contains custom-made ImageJ script used to analyze intensities in manually picked ROIs.
DOI: https://doi.org/10.7554/eLife.46629.009

though the sustained release of neurotransmitter is compromised, perhaps by the reduced number of SVs/bouton (*Figure 2B and C*) and/or their efficient recycling.

To test this hypothesis, we performed two types of experiments. Initially, immunocytochemistry was used to measure the total pool of SVs by staining neuronal cultures for the SV protein Synaptophysin. This revealed a reduction of Synaptophysin content per bouton (*Figure 3F and G*), indicating that SV loss (as seen in EM micrographs) (*Figure 2A and B*) is a putative contributor to the reduced functionality of $Pclo^{gt/gt}$ neurons. Next, we performed FM-dye uptake experiments (*Smith and Betz, 1996*) to determine the size of the total recycling pool (TRP) of SVs. Here, $Pclo^{gt/gt}$ neurons displayed a 30% reduction in FM4-64 intensity, consistent with a smaller TRP of SVs (*Figure 3H and I*). Finally, to assess how efficiently SVs are recycled and reused during high frequency stimulation, we determined FM1-43 unloading kinetics. Remarkably, FM1-43 dye was released at a much slower rate from $Pclo^{gt/gt}$ than $Pclo^{wt/wt}$ boutons (*Figure 3J,K*). Moreover, the total amount of FM1-43 dye released after 90 s stimulation was significantly reduced in neurons lacking Piccolo (*Figure 3K*). As SV exocytosis is normal, these data indicate that the recycling of SVs must be compromised.

## Levels of endosome proteins are reduced at $Pclo^{gt/gt}$ synapses

Our initial results raised several questions: a) why is SV pool size smaller in boutons lacking Piccolo and b) why are FM unloading rates slowed? Of note, $Pclo^{gt/gt}$ presynaptic terminals do have increased numbers of endosome-like membranes (*Figure 2B and F*), which could be explained by defects in the recycling or reformation of SVs.

As an initial test of this hypothesis, we expressed GFP-Rab5 to monitor the presence of endocytic compartments (*Spang, 2009*; *Stenmark, 2009*) in $Pclo^{gt/gt}$ neurons. Interestingly, we observed an increase in the number of GFP-Rab5 positive puncta along axons (*Figure 4B and C*). To examine whether this was associated with a general up regulation of the endo-lysosomal pathway, we monitored the presence of GFP-Rab7, a marker for late endosomes (*Zerial and McBride, 2001*) (*Figure 4A*). Surprisingly, the number of GFP-Rab7 puncta along axons was reduced in $Pclo^{gt/gt}$ neurons (*Figure 4B and C*), indicating that the maturation of membranes within the endosome compartment from Rab5 positive early endosomes towards late Rab7 positive endosomes is affected by the loss of Piccolo (*Figure 4A*).

To gain further insights into which maturation step is possibly impaired (*Figure 4*), we also analyzed the number of EEA1 positive puncta along axons, a marker for early endosomes. EEA1 is a docking factor, which facilitates homotypic fusion between early endocytic vesicles, facilitating the formation of early endosomes (*Christoforidis et al., 1999*). Indeed, the number of EEA1 immuno-positive puncta along axons was reduced in $Pclo^{gt/gt}$ neurons (*Figure 4B and C*), further indicating that the loss of Piccolo alters early endocytic membrane trafficking.

As Piccolo is an AZ protein, we next analyzed whether the levels of Rab5, EEA1 and Rab7 were also changed at synapses. Note that more than 80% of the observed Rab5 puncta along axons co-localize with Synaptophysin, indicating that the major fraction of Rab5 puncta is synaptic (*Figure 4—figure supplement 2I*). Quantifying intensities at Synaptophysin positive boutons revealed that while Rab5 levels were not altered (*Figure 4D and G*), EEA1 levels were decreased at $Pclo^{gt/gt}$ synapses (*Figure 4*, E and H). The same is true for the synaptic levels of mChRab7; they are also reduced in $Pclo^{gt/gt}$ neurons (*Figure 4F and I*).

The recruitment of EEA1 to endocytic vesicles is dependent on active GTP bound Rab5 (*Mishra et al., 2010*; *Murray et al., 2016*; *Simonsen et al., 1998*). However, the synaptic levels of the GEF Rabex5, known to regulate Rab5 activity (*Horiuchi et al., 1997*) (*Figure 4A*), were not significantly decreased (*Figure 4—figure supplement 1 A and B*). Taken together, these data indicate that Piccolo normally contributes to the efficient recruitment of endosome proteins like EEA1 and Rab7 towards presynaptic boutons. Consistent with this concept, no differences in the levels of Rab5 and EEA1 were detected in the cell soma of $Pclo^{wt/wt}$ and $Pclo^{gt/gt}$ neurons (*Figure 4—figure supplement 1 B and C*). However, the levels of somatic Rabex5 were significantly reduced in $Pclo^{gt/gt}$ neurons (*Figure 4—figure supplement 1D*). Intriguingly, along dendrites the intensities of Rabex5 puncta are not altered upon the loss of Piccolo (*Figure 4—figure supplement 1 E and H*), although the intensities of dendritic Rab5 and EEA1 puncta are increased in $Pclo^{gt/gt}$ neurons (*Figure 4—figure supplement 1 F and G*).

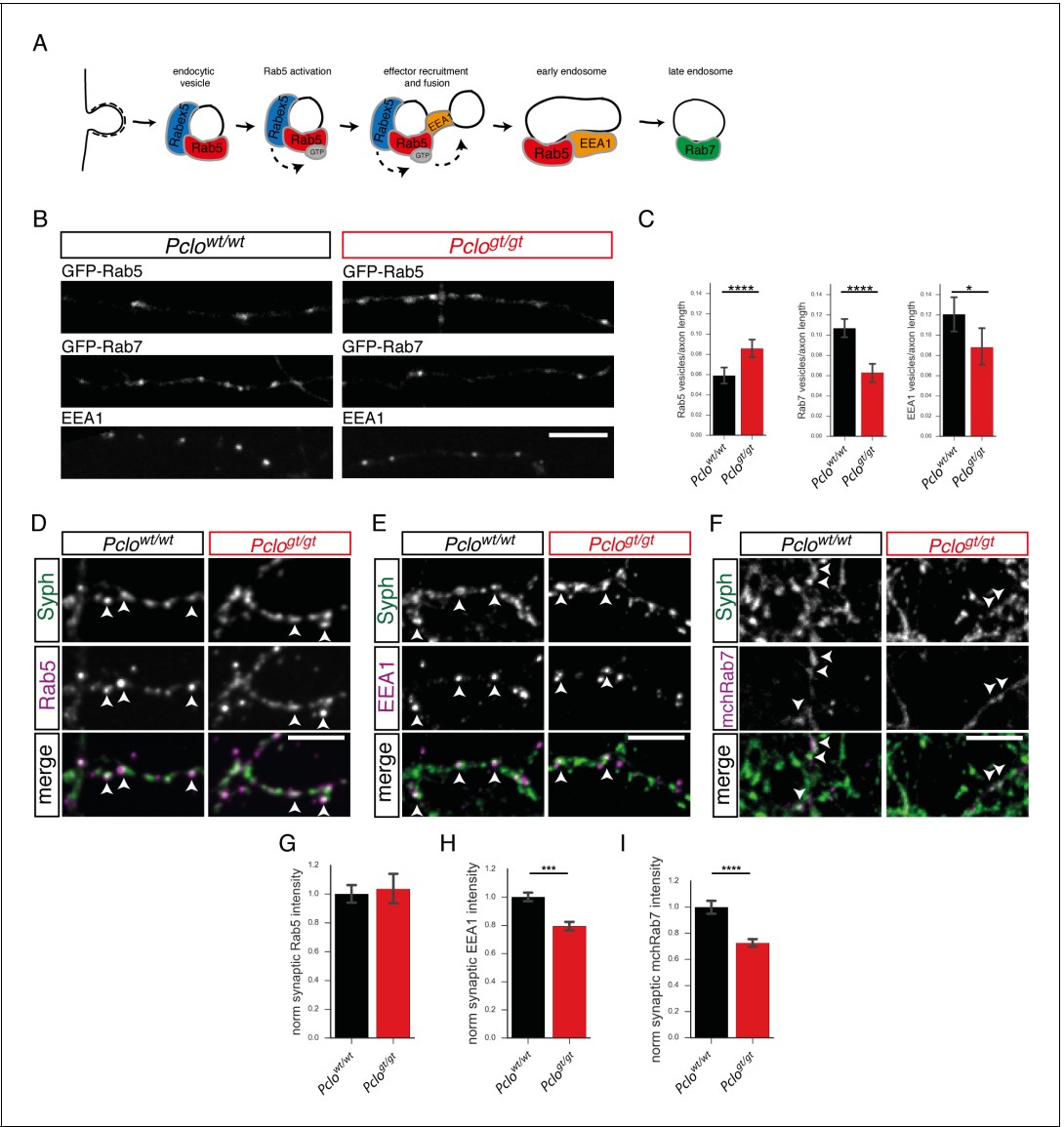

**Figure 4.** Levels of endosome proteins are reduced at synapses lacking Piccolo. (**A**) Schematic of SV membrane trafficking, illustrating when Rab5, EEA1 and Rab7 become associated with endosomal membranes. (**B**) Images of axon segments expressing GFP-Rab5 or GFP-Rab7 or immuno-stained with antibodies against EEA1. (**C**) Quantification of B. In *Pclo*<sup>wt/wt</sup> neurons, fewer GFP-Rab5 puncta are present per unit length of axon (*Pclo*<sup>wt/wt</sup> = 0.06 ± 0.004, n = 139 axon sections; *Pclo*<sup>gt/gt</sup> = 0.08 ± 0.004, n = 143 axon sections; three independent experiments). In *Pclo*<sup>wt/wt</sup> neurons, more GFP-Rab7 puncta are present per unit length of axon (*Pclo*<sup>wt/wt</sup> = 0.11 ± 0.004, n = 139 axon sections; *Pclo*<sup>gt/gt</sup> = 0.06 ± 0.004, n = 142 axon sections; two independent experiments). In *Pclo*<sup>wt/wt</sup> neurons, more EEA1 positive puncta are present per unit length of axon compared to *Pclo*<sup>gt/gt</sup> neurons (*Pclo*<sup>wt/wt</sup> = 0.12 ± 0.01, n = 29 axon sections; *Pclo*<sup>gt/gt</sup> = 0.09 ± 0.004, n = 34 axon sections; two independent experiments). (**D–F**) Immunocytochemical stainings of hippocampal neurons for endosome proteins. (**D**) Rab5 is present at *Pclo*<sup>wt/wt</sup> and *Pclo*<sup>gt/gt</sup> synapses, no differences in intensities are detectable. (**E**) EEA1 intensities are significantly reduced in *Pclo*<sup>gt/gt</sup> synapses. (**F**) mChRab7 intensities are significantly reduced in *Pclo*<sup>gt/gt</sup> synapses. (**G–I**) Quantification of (**D–F**). (**G**) The levels of Rab5 are not altered between *Pclo*<sup>wt/wt</sup> and *Pclo*<sup>gt/gt</sup> synapses (*Pclo*<sup>wt/wt</sup> = 1 ± 0.03, n = 808 synapses; *Pclo*<sup>gt/gt</sup> = 1.04 ± 0.05, n = 773 synapses; three independent experiments). (**H**) EEA1 intensity is significantly reduced in *Pclo*<sup>gt/gt</sup> synapses (*Pclo*<sup>wt/wt</sup> = 1 ± 0.02, n = 4323 synapses; *Pclo*<sup>gt/gt</sup> = 0.79 ± 0.02, n = 3939 synapses; 10 independent experiments). (**I**) mChRab7 intensity is reduced in *Pclo*<sup>gt/gt</sup> synapses (*Pclo*<sup>wt/wt</sup> = 1 ± 0.03, n = 386 synapses; *Pclo*<sup>gt/gt</sup> = 0.72 ± 0.02, n = 525 synapses; three independent experiments). Scale bar represents 10 μm. Error bars in bar graph represent 95% confidence intervals. Numbers given represent mean ± SEM, Student's *t*-test. * denotes p<0.05, *** denotes p<0.001 and **** denotes p<0.0001.

DOI: https://doi.org/10.7554/eLife.46629.010

The following source data and figure supplements are available for figure 4:

**Source data 1.** This spreadsheet contains the normalized values used to generate the bar plots shown in *Figure 4C,H,I and J*.

DOI: https://doi.org/10.7554/eLife.46629.013

*Figure 4 continued on next page*

*Figure 4 continued*

**Figure supplement 1.** Levels of endosome proteins are not altered in the soma or along dendrites in *Pclo$^{gt/gt}$* neurons.
DOI: https://doi.org/10.7554/eLife.46629.011

**Figure supplement 2.** PI3P in Piccolo *Pclo$^{wt/wt}$* and *Pclo$^{gt/gt}$* synapses.
DOI: https://doi.org/10.7554/eLife.46629.012

## PI3P-positive endosome organelles are smaller at *Pclo$^{gt/gt}$* synapses

The phosphoinositide PI3P is one of the hallmark lipids of EEA1 positive early endosomes (*Schink et al., 2013*). The FYVE domain of Hrs specifically binds PI3P (*Komada and Soriano, 1999*), making it a specific marker to visualize early endosomes in cells (GFP-2x-FYVE) (*Figure 4—figure supplement 2A*). To examine whether reduced synaptic EEA1 levels in *Pclo$^{gt/gt}$* neurons could be due to low levels of PI3P, GFP-2x-FYVE was expressed in primary hippocampal neurons, where it labeled puncta along axons (*Figure 4—figure supplement 2B*). The numbers per axon length were not different between *Pclo$^{wt/wt}$* and *Pclo$^{gt/gt}$* neurons (*Figure 4—figure supplement 2B and C*), suggesting that PI3P-positive organelles form normally. This is further supported by the fact that also along dendrites and in the soma no differences in GFP-2x-FYVE levels are detectable (*Figure 4—figure supplement 1 A and E*).

However, to gain further insight into the size and distribution of PI3P-positive organelles at synapses, we performed super resolution microscopy on neurons expressing GFP-2x-FYVE. Using structure illumination microscopy (SIM), we observed that on average *Pclo$^{gt/gt}$* boutons (Bassoon and Synaptophysin positive) had smaller GFP-2x-FYVE organelles than *Pclo$^{wt/wt}$* boutons (*Figure 4—figure supplement 2G and H*). These data indicate an altered formation of early endosomes in *Pclo$^{gt/gt}$* boutons, as fewer, larger organelles representing early endosomes are present. This is consistent with the prevalence of small endocytic vesicles (60–100 nm) seen in electron micrographs of *Pclo$^{gt/gt}$* boutons (*Figure 2*).

## Fewer endosome proteins are recruited to PI3P-positive organelles in *Pclo$^{gt/gt}$* neurons

A possible explanation for smaller PI3P-positive organelles in *Pclo$^{gt/gt}$* boutons is that the transition between small early endocytic vesicles and larger early endosomes is attenuated. To test this hypothesis, we analyzed the recruitment of two endogenous endosome markers, Rab5 and EEA1, towards GFP-2x-FYVE puncta. Here, we analyzed the fraction of GFP-2x-FYVE/Rab5 double positive puncta as a measure for early endocytic vesicles, and the fraction of GFP-2x-FYVE/Rab5/EEA1 triple positive puncta as a measure for early endosomes (*Figure 5A*). For technical reasons this analysis was performed along axons without an additional co-staining for synapses. Of note, co-localization studies with Synaptophysin demonstrated that at least 65% of GFP-2x-FYVE positive puncta along axons were synaptic (*Figure 4—figure supplement 2D and E*), a concept previously reported at the *Drosophila* NMJ (*Wucherpfennig et al., 2003*). Therefore our analysis represents a mixture of PI3P-positive organelles in- and outside synapses.

In *Pclo$^{gt/gt}$* axons, the amount of endogenous EEA1 as well as Rab5 at GFP-2x-FYVE puncta was significantly decreased (*Figure 5B,C,D*). Of note, although the intensity of Rab5 at GFP-2x-FYVE puncta was decreased in *Pclo$^{gt/gt}$* neurons, the overall fraction of puncta that were double positive for GFP-2x-FYVE and Rab5 was not significantly altered in comparison to *Pclo$^{wt/wt}$* neurons (*Figure 5E*). In contrast, the fraction of early endosomes (GFP-2x-FYVE/Rab5/EEA1) was reduced by about 70% (*Figure 5F*). Taken together, these data indicate that the recruitment of EEA1 towards PI3P-positive organelles is affected by the loss of Piccolo, slowing the maturation of early endocytic vesicles into early endosomes.

As EEA1 recruitment to early endocytic structures depends on active Rab5 (*Mishra et al., 2010*; *Murray et al., 2016*; *Simonsen et al., 1998*) and PI3P, which is not altered in *Pclo$^{gt/gt}$* neurons (*Figure 4—figure supplement 2F*), it remains possible that Rab5 activation/activity is decreased. In fact levels of Rabex5 at GFP-2x-FYVE puncta were decreased in *Pclo$^{gt/gt}$* neurons (*Figure 5—figure supplement 1C and D*). Notably, the fraction of GFP-2x-FYVE/Rabex5 double positive puncta is significantly increased in *Pclo$^{gt/gt}$* neurons (*Figure 5—figure supplement 1E*). In contrast, those that are triple positive for GFP-2x-FYVE, Rabex5 and Rab5 are reduced by 40% (*Figure 5—figure*

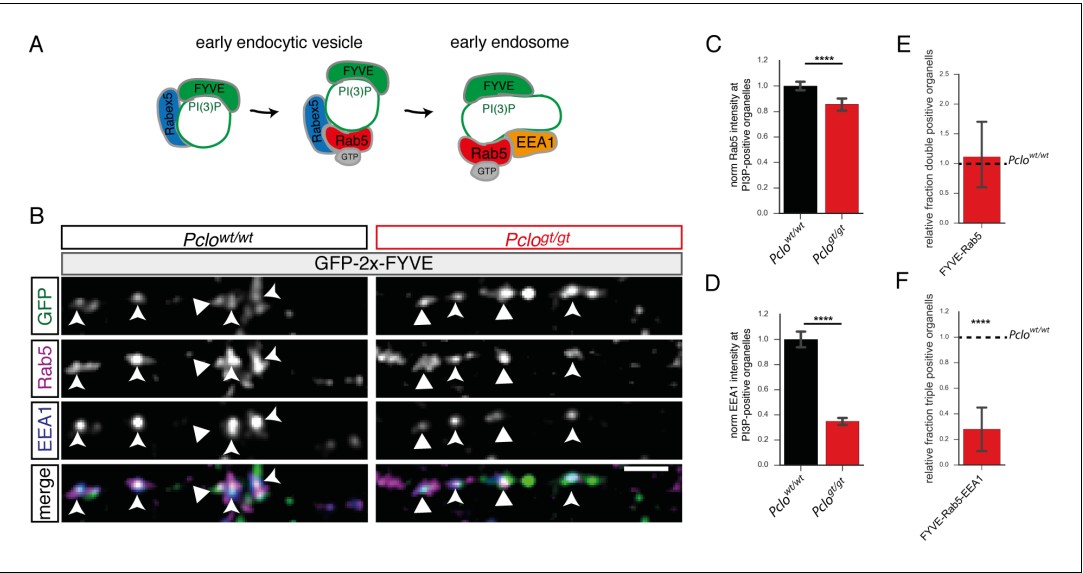

**Figure 5.** Fewer endosome proteins are recruited towards PI3P-positive organelles along axons lacking Piccolo. (A) Schematic of early endocytic trafficking steps. After pinching off from the plasma membrane early endocytic vesicles undergo consecutive maturation steps. The lipid PI3P is generated and a stable complex consisting of Rab5 and its GEF Rabex5 is formed creating a pool of active Rab5. This step is necessary to recruit EEA1 and form early endosomes. (B) Images depicting Rab5 and EEA1 intensities at GFP-2x-FYVE organelles along axons in *Pclo*<sup>gt/gt</sup> vs *Pclo*<sup>wt/wt</sup> neurons. (C–F) Quantification of B. (C) The levels of Rab5 at PI3P-positive organelles are decreased (*Pclo*<sup>wt/wt</sup> = 1 ± 0.02, n = 1645 puncta; *Pclo*<sup>gt/gt</sup> = 0.81 ± 0.02, n = 1233 puncta; six independent experiments). (D) The amount of EEA1 at endosome membranes is reduced (*Pclo*<sup>wt/wt</sup> = 1 ± 0.04, n = 1634 puncta; *Pclo*<sup>gt/gt</sup> = 0.37 ± 0.02, n = 1169 puncta; six independent experiments). (E) Quantification of double positive compartments along axons. The fraction of GFP-2x-FYVE/Rab5 is not altered (*Pclo*<sup>gt/gt</sup> = 0.97 ± 0.33, n = 5 independent experiments). (F) The relative percentage of GFP-2x-FYVE/Rab5/EEA1 positive vesicles is decreased in *Pclo*<sup>gt/gt</sup> neurons (GFP-2x-FYVE/Rab5/EEA1: *Pclo*<sup>gt/gt</sup> = 0.28 ± 0.10, n = 6 independent experiments). Scale bars represent 10 μm. Error bars in bar graph represent 95% confidence intervals. Numbers given represent mean ± SEM, Student's *t* -test. * denotes p<0.05, ** denotes p<0.01, *** denotes p<0.001 and **** denotes p<0.0001.

DOI: https://doi.org/10.7554/eLife.46629.014

The following source data and figure supplement are available for figure 5:

**Source data 1.** This spreadsheet contains the normalized values used to generate the bar plots shown in *Figure 5C,D,E and F*.

DOI: https://doi.org/10.7554/eLife.46629.016

**Figure supplement 1.** Synaptic Rabex5 levels as well as Rabex5 amounts at PI3P-positive organelles are reduced in neurons lacking Piccolo.

DOI: https://doi.org/10.7554/eLife.46629.015

*supplement 1F*). Together, this suggests that the reduced recruitment of EEA1 to Rab5/PI3P early endocytic organelles could be due to an altered Rab5 activation/activity.

## GDP-locked Rab5 (Rab5<sup>S34N</sup>) expression in *Pclo*<sup>wt/wt</sup> neurons decreases early endosomes and SV cycling to similar levels seen in *Pclo*<sup>gt/gt</sup> neurons

To probe this hypothesis, we expressed an inactive GDP-locked version of Rab5 (Rab5<sup>S34N</sup>) in *Pclo*<sup>wt/wt</sup> neurons, analyzing whether it can mimic the *Pclo*<sup>gt/gt</sup> endosome phenotype. Interestingly, Synaptophysin levels in neurons expressing Rab5<sup>S34N</sup> were decreased (*Figure 6A and C*), which was similar in magnitude to that observed in *Pclo*<sup>gt/gt</sup> neurons (*Figure 3F and G*). Furthermore the presence of Rab5<sup>S34N</sup> in *Pclo*<sup>wt/wt</sup> neurons led to lower levels of EEA1 at GFP-2x-FYVE puncta as well as a smaller fraction of GFP-2x-FYVE/Rab5/EEA1 triple positive early endosomes (*Figure 6B,E,G*). However, Rab5<sup>S34N</sup> expression did not alter the amount of Rab5 available at GFP-2x-FYVE puncta, though it significantly increased the fraction of GFP-2x-FYVE/Rab5 double positive organelles (*Figure 6B,D,F*). In addition, we observed that the FM1-43 uptake efficiency during 60 mM KCl stimulation was reduced by about 20% in neurons expressing Rab5<sup>S34N</sup> (*Figure 6H and I*), representing a smaller TRP of vesicles. This is consistent with our observations in *Pclo*<sup>gt/gt</sup> neurons (*Figure 3H and I*). The same was true for FM1-43 unloading kinetics during a 10 Hz/900AP train. Here, as in *Pclo*<sup>gt/gt</sup> neurons, less FM1-43 dye was released after 90 s stimulation in *Pclo*<sup>wt/wt</sup> neurons expressing

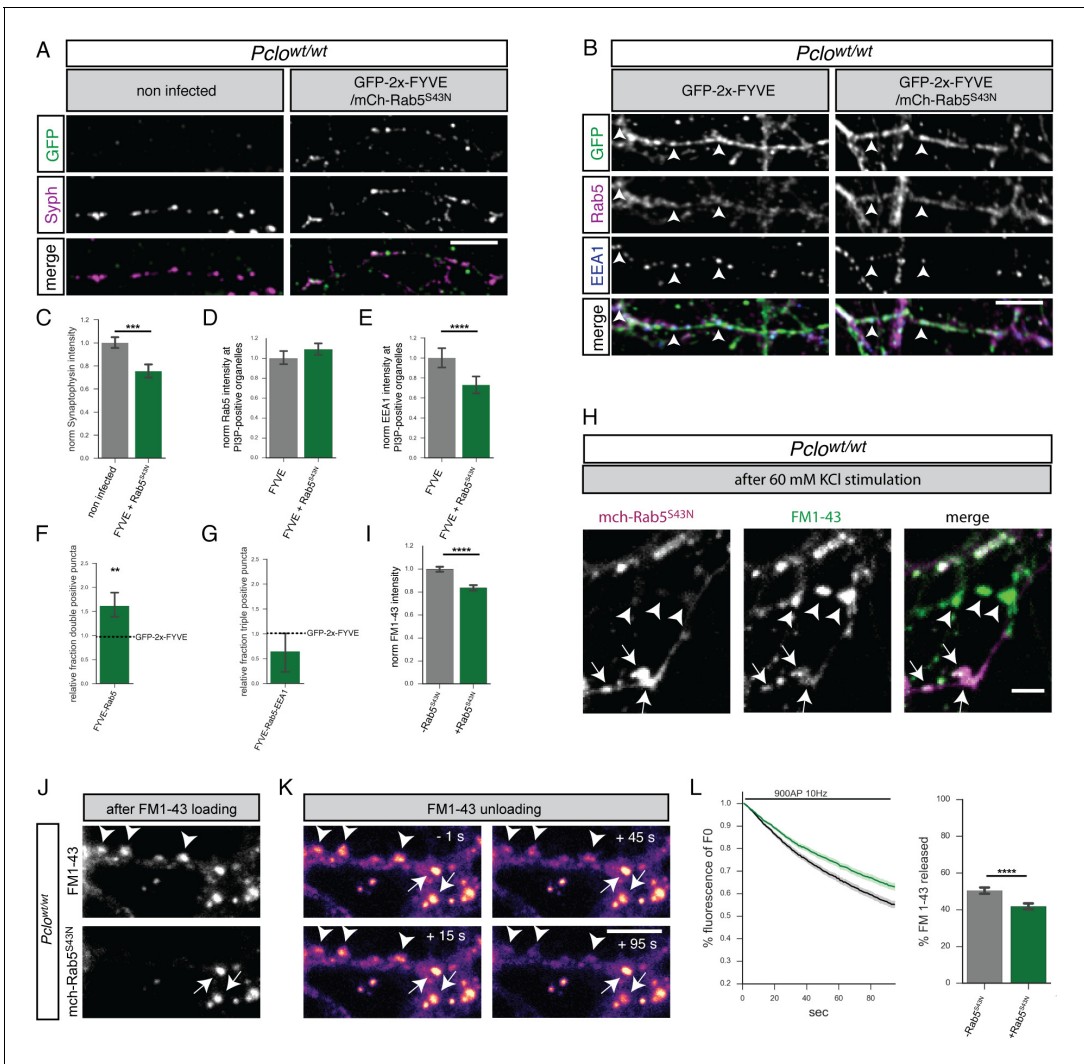

**Figure 6.** GDP-locked Rab5 (Rab5$^{S34N}$) expression in *Pclo$^{wt/wt}$* neurons decreases Synaptophysin levels and the number of early endosomes. (**A**) Images of *Pclo$^{wt/wt}$* neurons expressing GFP-2x-FYVE and mCh-Rab5$^{S34N}$ immuno-stained with Synaptophysin antibodies. (**B**) Images of *Pclo$^{wt/wt}$* neurons expressing GFP-2x-FYVE and mCh-Rab5$^{S34N}$ immuno-stained with GFP, Rab5 and EEA1 antibodies. (**C**) Quantification of (**A**). Synaptophysin levels are reduced in neurons expressing Rab5$^{S34N}$ (*Pclo$^{wt/wt}$* = 1 ± 0.02, n = 204 synapses; *Pclo$^{wt/wt}$* + mCh-Rab5$^{S34N}$ = 0.75 ± 0.03, n = 139 synapses; two independent experiments). (**D–G**) Quantification of B. (**D**) mCh-Rab5$^{S34N}$ expression slightly increases Rab5 levels at PI3P-positive organelles (*Pclo$^{wt/wt}$* = 1 ± 0.03, n = 930 puncta; *Pclo$^{wt/wt}$* + mCh-Rab5$^{S34N}$ = 1.09 ± 0.03, n = 612 puncta; three independent experiments). (**E**) EEA1 levels at PI3P-positive organelles decrease upon mCh-Rab5$^{S34N}$ expression (*Pclo$^{wt/wt}$* = 1 ± 0.05, n = 930 puncta; *Pclo$^{wt/wt}$* + mCh-Rab5$^{S34N}$ = 0.73 ± 0.04, n = 613; three independent experiments). (**F**) More FYVE/Rab5 positive organelles are present in mCh-Rab5$^{S34N}$ expressing neurons (*Pclo$^{wt/wt}$* = 1 ± 0; *Pclo$^{wt/wt}$* + mCh-Rab5$^{S34N}$ = 1.614 ± 0.15; n = 4 independent experiments). (**G**) Fewer FYVE/Rab5/EEA1 triple positive organelles are present in mCh-Rab5$^{S34N}$ expressing neurons (*Pclo$^{wt/wt}$* = 1 ± 0; *Pclo$^{wt/wt}$* + mCh-Rab5$^{S34N}$ = 0.64 ± 0.22; n = 4 independent experiments). (**H**) Representative images depicting FM1-43 uptake efficiency after KCl stimulation in boutons either expressing Rab5$^{S34N}$ (arrow) or not (arrowhead). (**I**) Quantification of (**H**). Less FM1-43 dye is taken up during KCl stimulation in the presence of mCh-Rab5$^{S34N}$ (*Pclo$^{wt/wt}$* = 1 ± 0.01, n = 732 puncta; *Pclo$^{wt/wt}$* + mCh-Rab5$^{S34N}$ = 0.84 ± 0.01, n = 866 puncta; three independent experiments). (**J**) Images showing FM1-43 levels in *Pclo$^{wt/wt}$* neurons expressing mCh-Rab5$^{S34N}$ after 900AP 10 Hz field stimulation. (**K**) Representative images showing FM1-43 unloading kinetics in synapses shown in (**J**) (note boutons expressing or not expressing mCh-Rab5$^{S34N}$ are labeled with arrows or arrowheads, respectively). (**L**) Quantification of (**K**). Neurons expressing mCh-Rab5$^{S34N}$ (green trace) release less FM1-43 dye with slower kinetics compared to Pclo$^{wt/wt}$ neurons (black trace) (*Pclo$^{wt/wt}$* = 50.43 ± 0.8471, n = 293 synapses; *Pclo$^{wt/wt}$* + mCh-Rab5$^{S34N}$ = 41.79 ± 0.8052, n = 421 synapses, three independen experiments). Scale bars in A and B represent 10 µm, in H and K 5 µm, Error bars in bar graph represent 95% confidence intervals. Numbers given represent mean ± SEM Students's *t* - test. * denotes p<0.05, ** denotes p<0.01, *** denotes p<0.001, **** denotes p<0.0001.

DOI: https://doi.org/10.7554/eLife.46629.017

The following source data is available for figure 6:

**Source data 1.** This spreadsheet contains the normalized values used to generate the bar plots shown in *Figure 6D,E,F,G,I and L*.

DOI: https://doi.org/10.7554/eLife.46629.018

Rab5$^{S34N}$ (**Figure 6L** right panel). In addition, the dye was released more slowly (**Figure 6J,K,L**). Of note, the impact of Rab5$^{S34N}$ on FM1-43 dye unloading kinetics is less pronounced than in boutons lacking Piccolo (**Figure 3K**). Nonetheless, the observed similarities between *Pclo$^{gt/gt}$* and *Pclo$^{wt/wt}$* neurons expressing Rab5$^{S34N}$ support our initial hypothesis that less Rab5 activation could be responsible for reduced early endosome numbers in *Pclo$^{gt/gt}$* neurons.

## GTPase deficient Rab5 (Rab5$^{Q79L}$) expression in *Pclo$^{gt/gt}$* neurons rescues EEA1 and synaptophysin levels back to *Pclo$^{wt/wt}$* amounts

As Rab5$^{S34N}$ expression in *Pclo$^{wt/wt}$* neurons mimics *Pclo$^{gt/gt}$* phenotypes, we next tested whether the expression of dominant active, GTPase deficient Rab5 (Rab5$^{Q79L}$) can vice versa rescue Piccolo loss of function phenotypes. We therefore expressed mCh-Rab5$^{Q79L}$ (**Stenmark et al., 1994**) together with GFP-2x-FYVE in *Pclo$^{wt/wt}$* and *Pclo$^{gt/gt}$* neurons. Analyzing Rab5 and EEA1 levels at GFP-2x-FYVE puncta revealed a significant rescue of endogenous Rab5 as well as EEA1 levels in *Pclo$^{gt/gt}$* neurons (**Figure 7A,B,C,D**). Remarkably, the expression of Rab5$^{Q79L}$ in *Pclo$^{wt/wt}$* neurons had the opposite effect, decreasing EEA1 levels (**Figure 7A and D**), though total Rab5 levels were only slightly altered (**Figure 7A and C**).

To assess whether Rab5$^{Q79L}$ also rescues SV pool size in *Pclo$^{gt/gt}$* neurons, mCh-Rab5$^{Q79L}$/GFP-2x-FYVE expressing neurons were immuno-stained for Synaptophysin. Synaptophysin levels increased upon the presence of dominant active Rab5$^{Q79L}$ (**Figure 7E and F**), indicating that the loss of SVs in *Pclo$^{gt/gt}$* boutons is linked to altered Rab5 activity.

To assess whether the presence of Rab5$^{Q79L}$ can also rescue the smaller TRP of SVs and impaired SV cycling seen in *Pclo$^{gt/gt}$* neurons, we performed FM1-43 dye uptake and unloading experiments (**Figure 8A**). Here, we found that in *Pclo$^{gt/gt}$* synapses harboring Rab5$^{Q79L}$ the amount of FM1-43 dye after 60 mM KCl stimulation was comparable or even higher than in *Pclo$^{wt/wt}$* neurons (**Figure 8A and B**). In contrast, the presence of Rab5$^{Q79L}$ did not alter the amount of FM1-43 dye taken up in *Pclo$^{wt/wt}$* neurons (**Figure 8A and B**). Surprisingly, Rab5$^{Q79L}$ expression in *Pclo$^{gt/gt}$* neurons did not restore FM1-43 unloading kinetics (**Figure 8C and D**). In fact, the presence of Rab5$^{Q79L}$ decreased FM1-43 unloading in *Pclo$^{wt/wt}$* neurons to a similar extent as observed in *Pclo$^{gt/gt}$* neurons (**Figure 8C and D**). The same is true for the total amount of FM1-43 dye that was released. Here, the reduced levels of FM1-43 in *Pclo$^{gt/gt}$* boutons were not restored. Furthermore, the portion that was released in *Pclo$^{wt/wt}$* neurons was also decreased (**Figure 8D**).

## Silencing synaptic activity affects synaptic levels of endosomal proteins

At synapses, endocytosis is associated with synaptic activity as part of the fusion and recycling of SVs. Given that cultured hippocampal neurons are intrinsically active (**Minerbi et al., 2009**), we considered the possibility that this activity creates a pool of Rab5 positive early endocytic vesicles in *Pclo$^{gt/gt}$* boutons that subsequently only poorly mature into early endosomes. To test this concept, hippocampal cultures were treated with Tetrodotoxin (TTX) for 24 hr and the synaptic levels of the endosome markers were examined (**Figure 8—figure supplement 1**). Interestingly, in *Pclo$^{wt/wt}$* neurons, the block of synaptic activity causes the levels of EEA1 to drop (**Figure 8—figure supplement 1D, F**). The levels of Rab5 and Rabex5 were only slightly altered (**Figure 8—figure supplement 1A, C, G, I**). This is consistent with a role of synaptic activity in the formation of early endosomes. However, silencing synapses in *Pclo$^{gt/gt}$* neurons had no effect on EEA1 levels, and only minor effects on the levels of Rabex5, Rab5 (**Figure 8—figure supplement 1B, C, E, F, H, I**). These data support the concept that proteins involved in the formation of presynaptic early endosomes are recruited into synapses in response to synaptic activity.

## Synaptic levels of prenylated rab acceptor protein 1 (Pra1) are reduced in *Pclo$^{gt/gt}$* neurons

We next asked the question how Piccolo could be linked to the endosome pathway within presynaptic terminals? Piccolo is best known for its role for the dynamic assembly of presynaptic F-actin (**Waites et al., 2011**). Therefore, we first tested whether the observed endosome phenotype is attributable to Piccolo's essential role in F-actin assembly. Here, we examined the effects of drugs, that either block (Latrunculin A) or enhance (Jasplakinolide) the assembly of F-actin, on the recruitment of Rab5 and EEA1 towards GFP-2x-FYVE punta. We found that the addition of Latrunculin

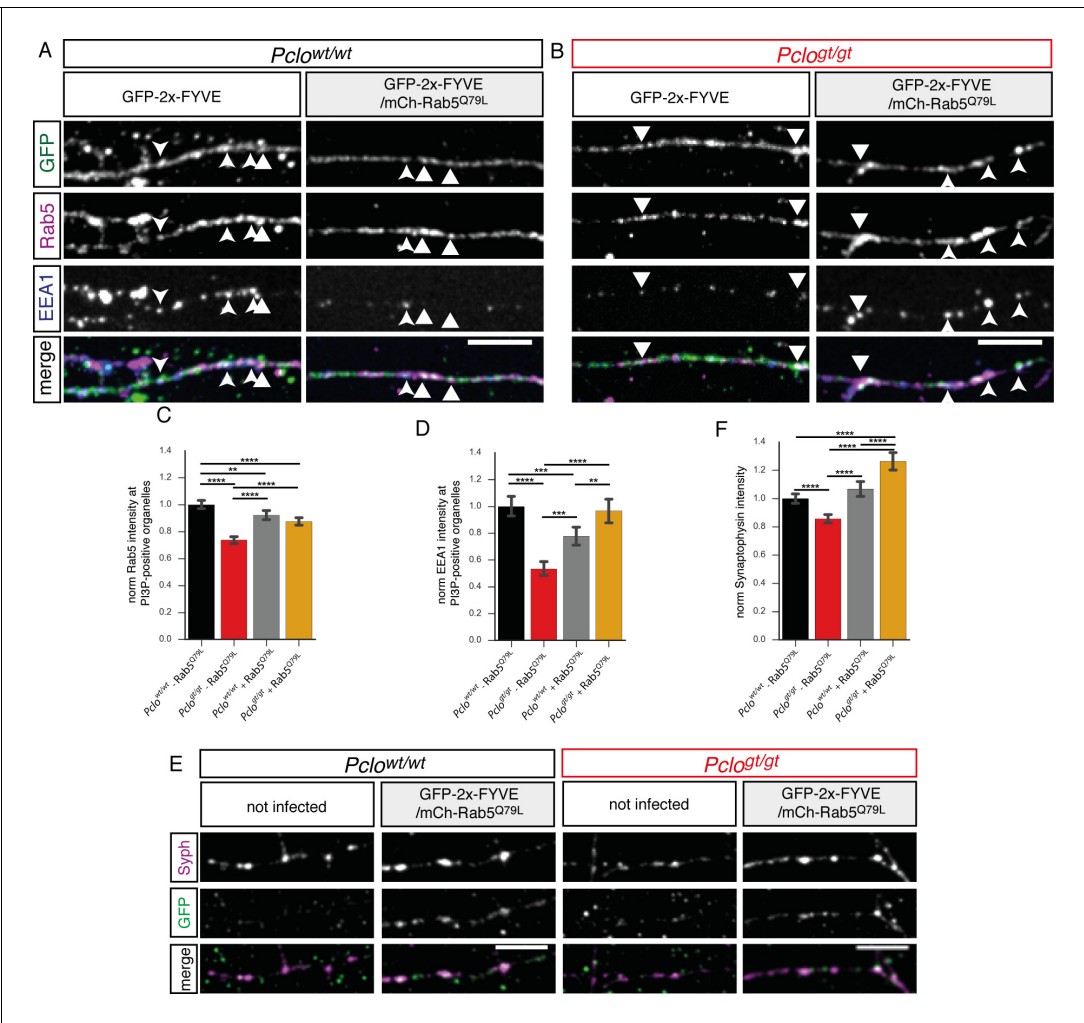

**Figure 7.** Expression of GTPase deficient Rab5 (Rab5$^{Q79L}$) in *Pclo$^{gt/gt}$* neurons rescues EEA1 levels at PI3P containing membranes and Synaptophysin back to *Pclo$^{wt/wt}$* levels. (**A–B**) Rab5 and EEA1 levels at GFP-2x-FYVE organelles along axons in *Pclo$^{wt/wt}$* (**A**) and *Pclo$^{gt/gt}$* (**B**) neurons and those expressing mCh-Rab5$^{Q79L}$. (**B**) In *Pclo$^{gt/gt}$* neurons, mCh-Rab5$^{Q79L}$ expression increases Rab5 and EEA1 level at GFP-2x-FYVE organelles towards *Pclo$^{wt/wt}$* levels. (**C–D**) Quantification of (**A and B**). (**C**) mCh-Rab5$^{Q79L}$ expression in *Pclo$^{wt/wt}$* neurons slightly reduces Rab5 levels at GFP-2x-FYVE membranes (*Pclo$^{wt/wt}$* = 1 ± 0.02, n = 1066 puncta; *Pclo$^{wt/wt}$* $_{(mCh-Rab5}$$^{Q79L}$$_)$=0.93 ± 0.02, n = 979 puncta; three independent experiments). mCh-Rab5$^{Q79L}$ expression in *Pclo$^{gt/gt}$* neurons increases Rab5 levels at PI3P-positive organelles (*Pclo$^{gt/gt}$* = 0.74 ± 0.01, n = 840 puncta; *Pclo$^{gt/gt}$* $_{(mCh-Rab5}$$^{Q79L}$$_)$=0.87 ± 0.01, n = 1185 puncta; three independent experiments). (**D**) In *Pclo$^{wt/wt}$* neurons mCh-Rab5$^{Q79L}$ expression causes EEA1 levels to drop (*Pclo$^{wt/wt}$* = 1 ± 0.04, n = 1066 puncta; *Pclo$^{wt/wt}$* $_{(mCh-Rab5}$$^{Q79L}$$_)$=0.77 ± 0.04, n = 979 puncta; three independent experiments). mCh-Rab5$^{Q79L}$ expression rescues EEA1 levels in *Pclo$^{gt/gt}$* neurons (*Pclo$^{gt/gt}$* = 0.53 ± 0.03, n = 840 puncta; *Pclo$^{gt/gt}$* $_{(mCh-Rab5}$$^{Q79L}$$_)$=0.97 ± 0.05, n = 1185 puncta; three independent experiments). (**E**) mCh-Rab5$^{Q79L}$ expression in *Pclo$^{wt/wt}$* neurons increases Synaptophysin intensities. (**F**) Quantification of (**E**). Synaptophysin puncta intensity slightly increases upon mCh-Rab5$^{Q79L}$ expression in *Pclo$^{wt/wt}$* neurons. In *Pclo$^{gt/gt}$* neurons, mCh-Rab5$^{Q79L}$ expression rescues Synaptophysin levels higher than *Pclo$^{wt/wt}$* levels (*Pclo$^{wt/wt}$* = 1 ± 0.02, n = 620 synapses; *Pclo$^{gt/gt}$* = 0.86 ± 0.02, n = 526 synapses; *Pclo$^{wt/wt}$* $_{(mCh-Rab5}$$^{Q79L}$$_)$=1.07 ± 0.03, n = 473 synapses; *Pclo$^{gt/gt}$* $_{(mCh-Rab5}$$^{Q79L}$$_)$=1.26 ± 0.03, n = 446 synapses; three independent experiment). Scale bars represent 10 µm. Error bars in bar graph represent 95% confidence intervals. Numbers given represent mean ± SEM. C, D and H ANOVA with Tukey multi comparison test. ** denotes p<0.01, *** denotes p<0.001 and **** denotes p<0.0001.

DOI: https://doi.org/10.7554/eLife.46629.019

The following source data is available for figure 7:

**Source data 1.** This spreadsheet contains the normalized values used to generate the bar plots shown in *Figure 7C,D and F*.
DOI: https://doi.org/10.7554/eLife.46629.020

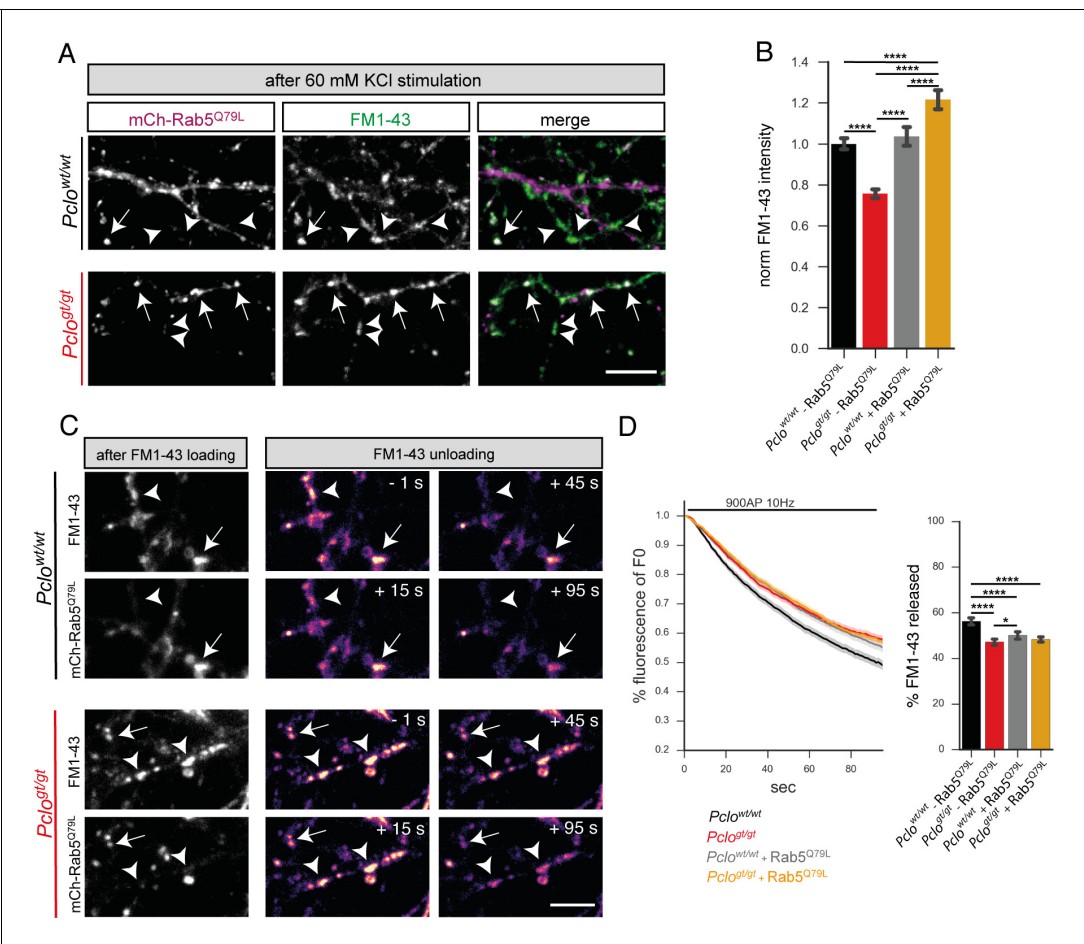

**Figure 8.** Expression of GTPase deficient Rab5 (Rab5$^{Q79L}$) in $Pclo^{gt/gt}$ neurons rescues the size of the total recycling pool of vesicles but not FM1-43 unloading kinetics. (**A**) Representative images showing FM1-43 levels in $Pclo^{wt/wt}$ and $Pclo^{gt/gt}$ boutons either positive (arrow) or negative (arrowhead) for mCh-Rab5$^{Q79L}$ after 60 mM KCl stimulation. (**B**) Quantification of (**A**). ($Pclo^{wt/wt}$ = 1 ± 0.01, n = 528 puncta; $Pclo^{gt/gt}$ = 0.76 ± 0.01, n = 654 puncta; $Pclo^{wt/wt}$ $_{(mCh-Rab5}$$^{Q79L}$$_{)}$=1.04 ± 0.02, n = 304 puncta; $Pclo^{gt/gt}$ $_{(mCh-Rab5}$$^{Q79L}$$_{)}$=1.22 ± 0.02, n = 337 puncta; three independent experiments). (**C**) Left: Images depicting FM1-43 intensities in $Pclo^{wt/wt}$ and $Pclo^{gt/gt}$ boutons either positive (arrow) or negative (arrowhead) for mCh-Rab5$^{Q79L}$. Right: Images depicting FM1-43 unloading kinetics in $Pclo^{wt/wt}$ and $Pclo^{gt/gt}$ boutons either positive (arrow) or negative (arrowhead) for mCh-Rab5$^{Q79L}$. (**D**) Quantification of (**C**). The presence of mCh-Rab5$^{Q79L}$ is not sufficient to rescue slowed FM1-43 unloading kinetics and total amount of FM1-43 dye released in $Pclo^{gt/gt}$ boutons ($Pclo^{gt/gt}$ = 47.24 ± 0.68, n = 361 synapses; $Pclo^{gt/gt}$$_{(Rab5}$$^{Q79L}$$_{)}$=48.33 ± 0.53, n = 479 synapses; four independent experiments). In $Pclo^{wt/wt}$ boutons expressing mCh-Rab5$^{Q79L}$, FM1-43 unloading kinetics are slowed and total amounts of dye released are reduced similar to what is observed in $Pclo^{gt/gt}$ boutons ($Pclo^{wt/wt}$ = 56.28 ± 0.81, n = 301 synapses; $Pclo^{wt/wt}$$_{(Rab5}$$^{Q79L}$$_{)}$=50.15 ± 0.85, n = 342 synapses; four independent experiments). Scale bars represent 5 μm. Error bars in bar graph represent 95% confidence intervals. Numbers given represent mean ± SEM. ANOVA with Tukey multi comparison test. * denotes p<0.05, **** denotes p<0.0001.

DOI: https://doi.org/10.7554/eLife.46629.021

The following source data and figure supplements are available for figure 8:

**Source data 1.** This spreadsheet contains the normalized values used to generate the bar plots shown in *Figure 8B and D*.

DOI: https://doi.org/10.7554/eLife.46629.024

**Figure supplement 1.** Silencing synaptic activity affects synaptic levels of endosome proteins in $Pclo^{wt/wt}$ and $Pclo^{gt/gt}$ neurons.

DOI: https://doi.org/10.7554/eLife.46629.022

**Figure supplement 2.** Deficiencies in F-actin assembly do not contribute to the endosome phenotype seen in boutons lacking Piccolo.

DOI: https://doi.org/10.7554/eLife.46629.023

significantly enhanced the levels of Rab5, but not EEA1 (*Figure 8—figure supplement 2A*). Consistently, Jasplakinolide was not able to enhance Rab5 and or EEA1 levels at GFP-2x-FYVE punta in boutons lacking Piccolo (*Figure 8—figure supplement 2B*), indicating that Piccolo's role in F-actin

assembly is not important for the formation of early endosomes. Hence the phenotype we observe must be due to a different function of Piccolo.

Interestingly Piccolo has been shown to also interact with Pra1 (*Fenster et al., 2000*), which is a GDI replacement factor. It is part of the Rab-GTPase activating/deactivating cycle, as it places the GTPase onto its target membrane (*Pfeffer and Aivazian, 2004*). It is thus tempting to speculate that $Pclo^{gt/gt}$ neurons lack synaptic Pra1, subsequently causing reduced Rab5 activation and impaired early endosome formation. To test this hypothesis, we analyzed the synaptic levels of Pra1 in $Pclo^{wt/wt}$ and $Pclo^{gt/gt}$ neurons. We found that Pra1 localized to presynaptic terminals, co-localizing with Synaptophysin in $Pclo^{wt/wt}$ neurons (*Figure 9A and B*). However, in $Pclo^{gt/gt}$ neurons synaptic Pra1 levels were reduced (*Figure 9A,B,C*). These data suggest that the reduced formation of

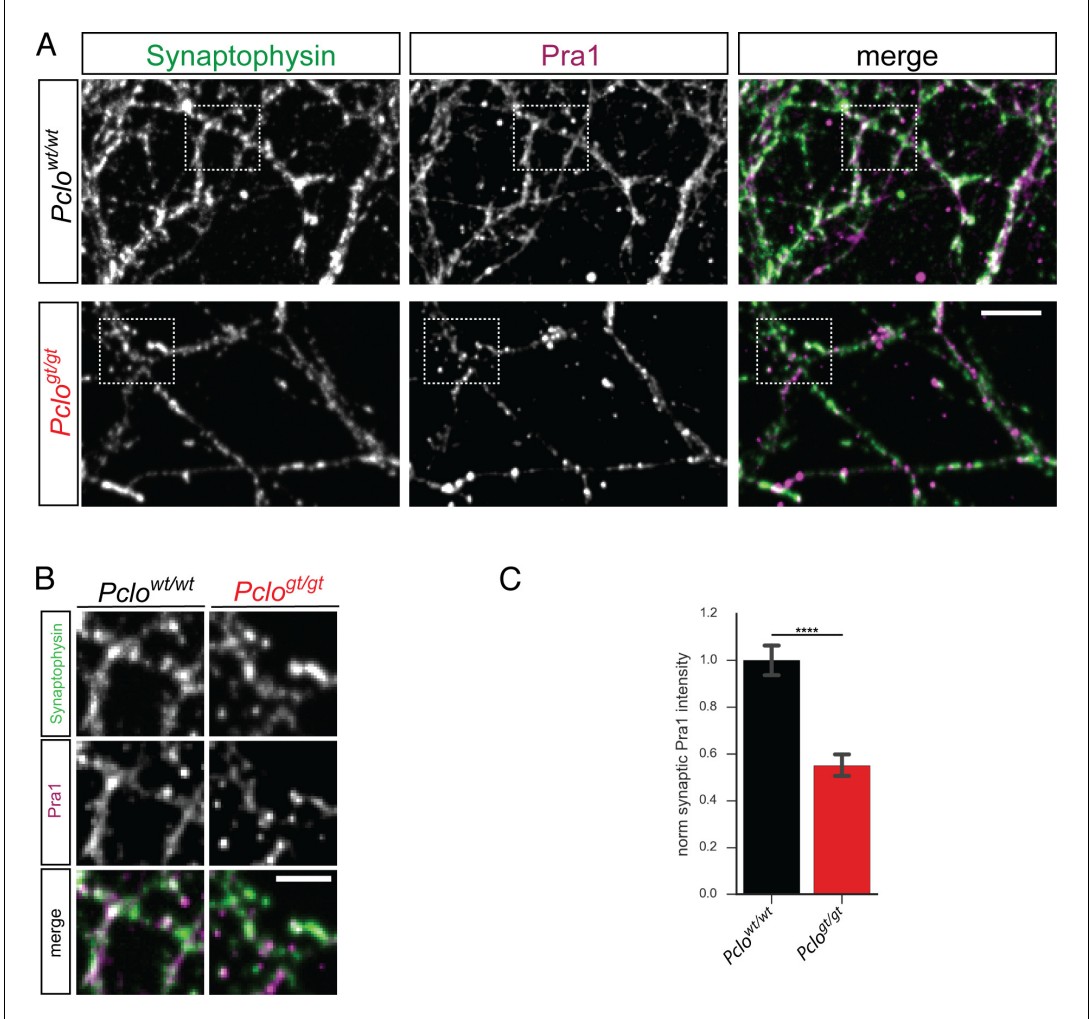

**Figure 9.** The loss of Piccolo leads to diminished levels of synaptic Pra1. (**A**) Images of $Pclo^{wt/wt}$ and $Pclo^{gt/gt}$ neurons stained with Synaptophysin and Pra1 antibodies. Pra1 is present at Synaptophysin positive synapses in $Pclo^{wt/wt}$ synapses but is reduced in $Pclo^{gt/gt}$ synapses. (**B**) Detail images of areas indicated in (**A**). (**C**) Quantitation of (**A**). Loss of Piccolo decreases Pra1 intensity at synapses compared to $Pclo^{wt/wt}$ synapses ($Pclo^{wt/wt}$ = 1 ± 0.03, n = 474 synapses; $Pclo^{gt/gt}$ = 0.55 ± 0.02, n = 723; three independent experiments). Scale bar represents 10 μm and 5 μm in zoom image, Error bars in bar graph represent 95% confidence intervals. Numbers given represent mean ± SEM, Student's $t$ –test. **** denotes p<0.0001.
DOI: https://doi.org/10.7554/eLife.46629.025

The following source data and figure supplement are available for figure 9:

**Source data 1.** This spreadsheet contains the normalized values used to generate the bar plots shown in *Figure 9C*.
DOI: https://doi.org/10.7554/eLife.46629.027
**Figure supplement 1.** Pclo-Znf1-mCh is nicely targeted to synapses.
DOI: https://doi.org/10.7554/eLife.46629.026

endosomal membranes in boutons lacking Piccolo is due to a reduced synaptic recruitment of its interaction partner Pra1.

## Expression of Pclo-Znf1 rescues synaptic Pra1 and EEA1 levels, synaptic vesicle pool size as well as SV cycling

As Piccolo interacts with Pra1 via its zinc fingers (*Fenster et al., 2000*), we hypothesized that this interaction is critical for its synaptic recruitment and subsequent role in Rab5 activation and early endosome formation. This suggests that the synaptic delivery of even one zinc finger could rescue the reduced synaptic levels of Pra1 and EEA1 in *Pclo$^{gt/gt}$* neurons. To test this hypothesis, we expressed mCh-tagged Znf1 of Piccolo (Pclo-Znf1-mCh) in *Pclo$^{wt/wt}$* and *Pclo$^{gt/gt}$* primary hippocampal neurons and stained for Synaptophysin, Pra1 and EEA1. First, we observed that more than 80% of Pclo-Znf1-mCh puncta co-localized with Synaptophysin in *Pclo$^{wt/wt}$* neurons (*Figure 9—figure supplement 1*), demonstrating it is mainly localized synaptically. Second, the expression of Pclo-Znf1-mCh rescued Synaptophysin levels to greater than wildtype levels in both *Pclo$^{gt/gt}$* and *Pclo$^{wt/wt}$* boutons (*Figure 10A and D*). Third, Pclo-Znf1-mCh over-expression also restored Pra1 levels at *Pclo$^{gt/gt}$* synapses to even higher levels than in *Pclo$^{wt/wt}$* neurons (*Figure 10B and E*). Fourth, *Pclo$^{wt/wt}$* synapses harboring Pclo-Znf1-mCh displayed a ~ 30% increase in the levels of synaptic Pra1 compared to synapses of un-infected *Pclo$^{wt/wt}$* neurons, indicating that the Piccolo Znf1 recruits Pra1 into presynaptic boutons. Also synaptic EEA1 levels in *Pclo$^{gt/gt}$* neurons were significantly increased in the presence Pclo-Znf1-mCh (*Figure 10C and F*), further indicating that Pclo-Znf1-mCh supports the accumulation of synaptic Pra1 and along with it EEA1.

Intriguingly, Pclo-Znf1-mCh expression in *Pclo* neurons also increased the ability of synapses to take up FM1-43 dye during 60 mM KCl stimulation. In *Pclo$^{wt/wt}$* as well as *Pclo$^{gt/gt}$* neurons, FM 1–43 levels were increased in the presence of Pclo-Znf1-mCh (*Figure 11A and B*). Considering FM1-43 unloading kinetics, the presence of Pclo-Znf1-mCh in *Pclo$^{wt/wt}$* neurons had only a minor effect. Also the total amount of FM1-43 dye that was released during the stimulation was only slightly affected (*Figure 11C and D*). In contrast, in *Pclo$^{gt/gt}$* neurons, Pclo-Znf1-mCh lead to a faster FM1-43 unloading rate compared to *Pclo$^{gt/gt}$* neurons, although the speed seen in *Pclo$^{wt/wt}$* neurons was not fully reached (*Figure 11C and D*). Finally, the total amount of dye released was increased (*Figure 11D*).

Taken together, these data support our central hypothesis that Piccolo via its zinc fingers and its binding partner Pra1 plays critical roles during the recycling and maintenance of SV pools, acting through the activation of Rab5 and EEA1 and the formation of early endosomes (*Figure 11E and F*).

## Discussion

Our study has shown that Piccolo plays a critical role in the activity dependent recycling of SVs. Specifically, we find that Piccolo loss of function reduces SV numbers as well as the recycling of SVs, without affecting the activity dependent docking and fusion of SVs. Mechanistically, we find that boutons lacking Piccolo accumulate endocytic vesicles that fail to mature into early endosomes due to an impaired Rab5 activation and recruitment of EEA1 towards PI3P-positive organelles. This appears to be a consequence of reduced presynaptic levels of Pra1, subsequently impacting total SV pool size and SV cycling dynamics, reflected in decreased FM-dye unloading kinetics. The observed phenotypes can be restored by the over-expression of Rab5$^{Q79L}$ as well as the Znf1 domain of Piccolo, which interacts with and restores synaptic Pra1 levels, SV pool sizes and SV cycling dynamics in Piccolo deficient synapses.

### Ultrastructural changes in boutons lacking Piccolo

Piccolo and its isoforms are encoded by a large 380 kb gene (*Pclo*) (*Fenster and Garner, 2002*). Using transposon mediated mutagenesis (*Izsvák et al., 2010*), a rat line was created that lacks high and low molecular weight isoforms of Piccolo. However not all isoforms are gone, indicating that the rat model is not representing a complete knockout model (*Figure 1*). It thus cannot entirely be excluded that truncated versions of Piccolo contribute to the here-observed phenotypes.

The loss of Piccolo does not affect the number of formed synapses (*Figure 1*), which is consistent with previous studies (*Leal-Ortiz et al., 2008*; *Mukherjee et al., 2010*). Though, the loss does lead to reduced levels of EEA1 and Synaptophysin in brain lysates (*Figure 1*) as well as synaptic levels of Synaptotagmin and VGlut1 (*Figure 1—figure supplement 1*). Consistently, our quantitative EM

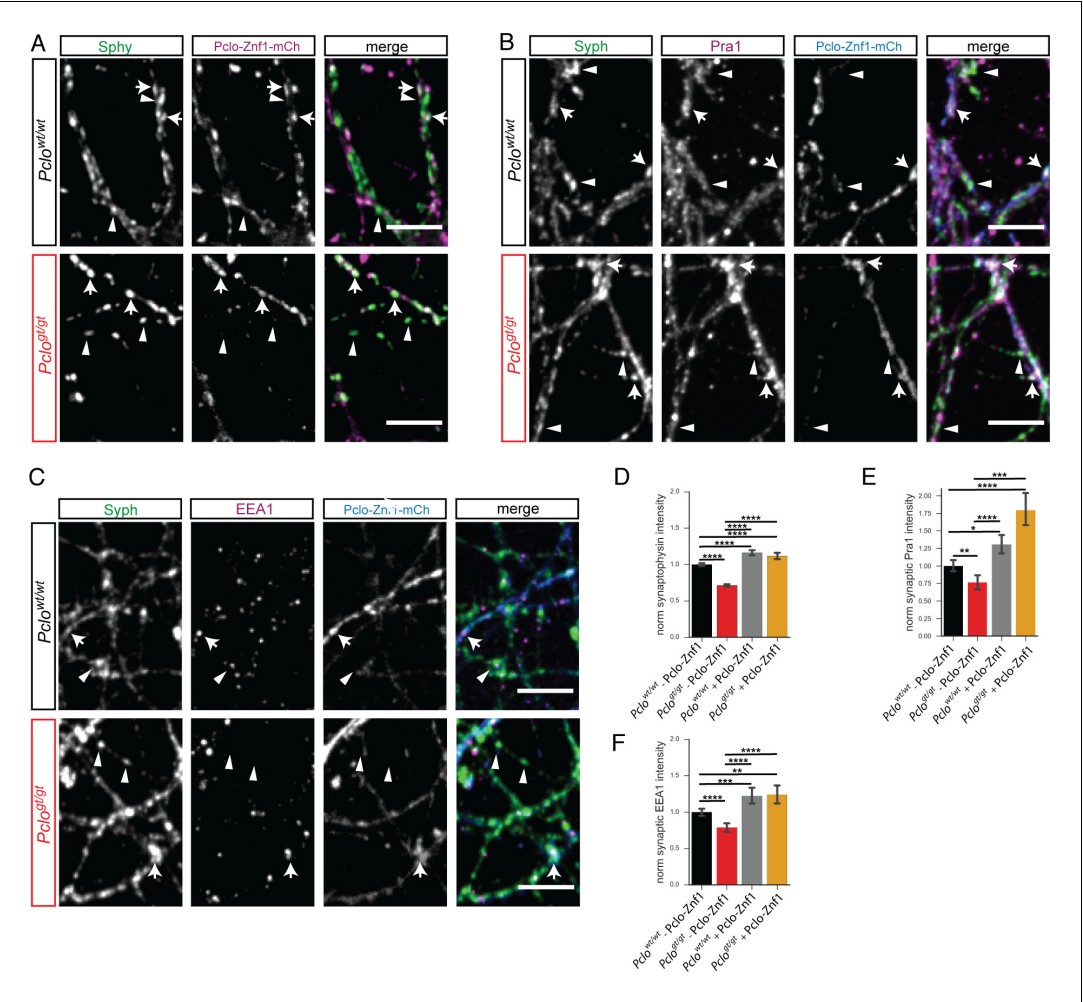

**Figure 10.** Expressing Pclo-Znf1-mCh rescues Pra1, EEA1 and Synaptophysin levels in $Pclo^{gt/gt}$ neurons. (A–C) Images of $Pclo^{wt/wt}$ and $Pclo^{gt/gt}$ neurons either expressing or not expressing Pclo-Znf1-mCh (arrows vs arrowheads) that were fixed and stained for Synaptophysin (A) and Pra1 (B) or EEA1 (C). (D) Quantification of (A). Pclo-Znf1-mCh slightly increases Synaptophysin levels in $Pclo^{wt/wt}$ neurons ($Pclo^{wt/wt}$ = 1 ± 0.01, n = 1497 synapses; $Pclo^{wt/wt}$ + Pclo-Znf1-mCh = 1.16 ± 0.02, n = 710; four independent experiments). In $Pclo^{gt/gt}$ boutons expressing Pclo-Znf1-mCh (arrows), Synaptophysin levels are increased compared to non-expressing boutons (arrowheads). ($Pclo^{gt/gt}$ = 0.71 ± 0.01, n = 1203; $Pclo^{gt/gt}$ + Pclo-Znf1-mCh = 1.12 ± 0.02, n = 590; four independent experiments). (E) Quantification of (B). Pclo-Znf1-mCh increases synaptic Pra1 levels in $Pclo^{wt/wt}$ neurons ($Pclo^{wt/wt}$ = 1 ± 0.04, n = 1136 synapses; $Pclo^{wt/wt}$ + Pclo-Znf1-mCh = 1.31 ± 0.07, n = 543; 3–5 independent experiments). Pclo-Znf1-mCh rescues and further increases synaptic Pra1 levels in $Pclo^{gt/gt}$ neurons ($Pclo^{gt/gt}$ = 0.76 ± 0.05, n = 1064; $Pclo^{gt/gt}$ + Pclo-Znf1-mCh = 1.80 ± 0.12, n = 445; 3–5 independent experiments). (F) Quantification of (C). Pclo-Znf1-mCh expression increases synaptic EEA1 levels in $Pclo^{wt/wt}$ neurons ($Pclo^{wt/wt}$ = 1 ± 0.03, n = 1652 synapses; $Pclo^{wt/wt}$ + Pclo-Znf1-mCh = 1.22 ± 0.05, n = 709; 3–5 independent experiments). Pclo-Znf1-mCh rescues and further increases synaptic EEA1 levels in $Pclo^{gt/gt}$ neurons ($Pclo^{gt/gt}$ = 0.79 ± 0.03, n = 1633; $Pclo^{gt/gt}$ + Pclo-Znf1-mCh = 1.24 ± 0.06, n = 464; 3–5 independent experiments). Scale bars represent 10 μm. Error bars in bar graph represent 95% confidence intervals. Numbers given represent mean ± SEM, ANOVA with Tukey multi comparison test. * denotes p<0.05, ** denotes p<0.01, *** denotes p<0.001 and **** denotes p<0.0001.
DOI: https://doi.org/10.7554/eLife.46629.028

The following source data is available for figure 10:

**Source data 1.** This spreadsheet contains the normalized values used to generate the bar plots shown in *Figure 10D,E and F*.
DOI: https://doi.org/10.7554/eLife.46629.029

analysis revealed a decreased number of SVs/bouton (*Figure 2*). Interestingly, the loss of SVs was associated with a concomitant increase in the number of small endocytic-like vesicles (~80 nm), suggesting that Piccolo is important for the recycling of SVs. It is worth noting that the deletion of exon 14 in the mouse $Pclo^{Ko/Ko}$ gene, which appears to only remove the longest Piccolo isoforms (*Mukherjee et al., 2010*; *Waites et al., 2011*), does not appear to affect the number of SVs/bouton

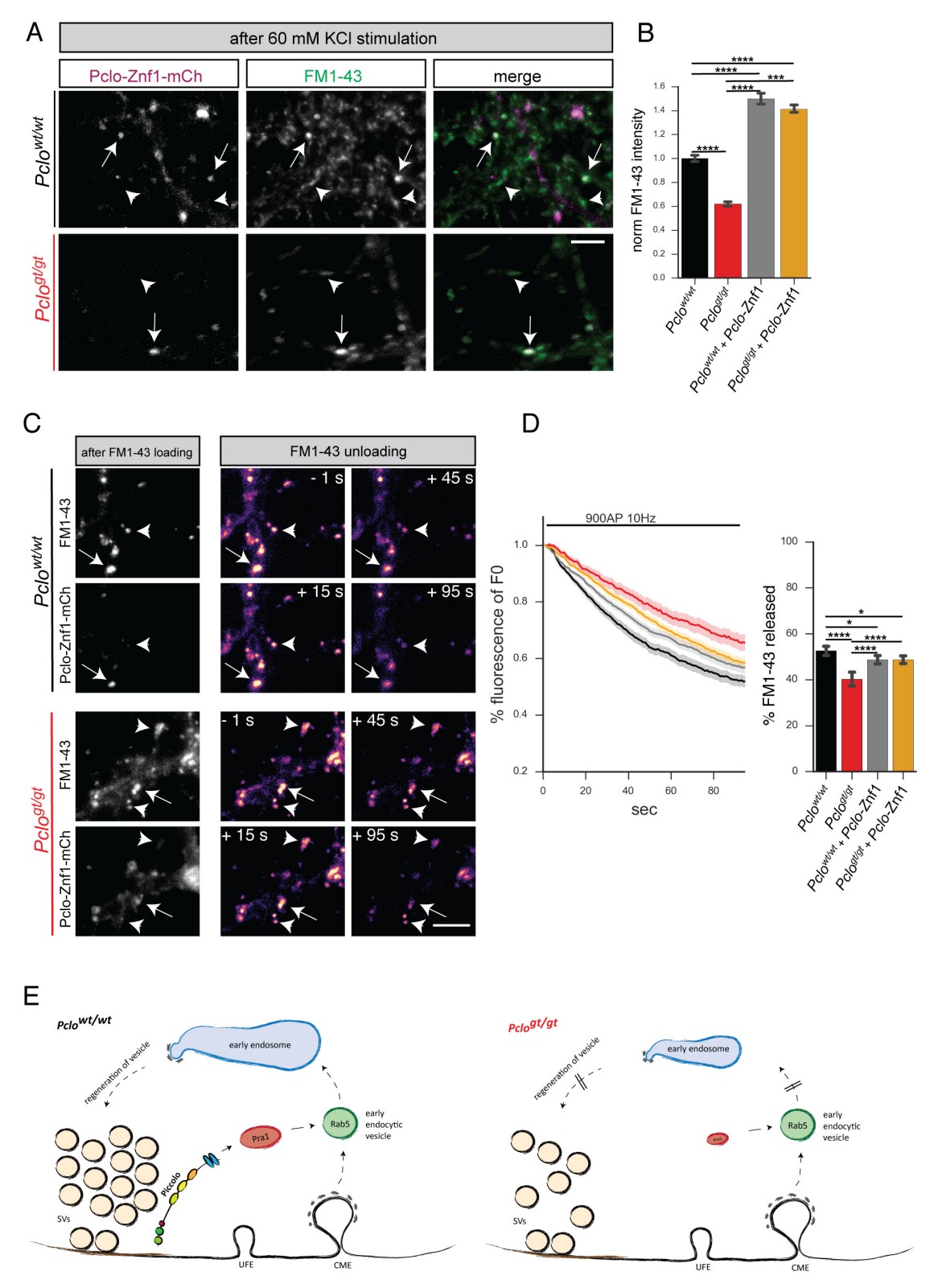

**Figure 11.** Expression of Pclo-Znf1-mCh in *Pclo^{gt/gt}* neurons rescues the size of the total recycling pool of vesicles and FM1-43 unloading kinetics. (**A**) Representative images showing FM1-43 levels in *Pclo^{wt/wt}* and *Pclo^{gt/gt}* boutons either positive for Pclo-Znf1-mCh (arrow) or negative (arrowhead) after 60 mM KCl stimulation. (**B**) Quantification of (**A**). (*Pclo^{wt/wt}* = 1 ± 0.01, n = 535 puncta; *Pclo^{gt/gt}* = 0.62 ± 0.01, n = 625 puncta; *Pclo^{wt/wt}* + Pclo-Znf1-mCh = 1.50 ± 0.02, n = 682 puncta; *Pclo^{gt/gt}* + Pclo-Znf1-mCh = 1.41 ± 0.01, n = 907 puncta; three independent experiments). (**C**) Left: Images depicting

*Figure 11 continued on next page*

*Figure 11 continued*

FM1-43 intensities in *Pclo*<sup>wt/wt</sup> and *Pclo*<sup>gt/gt</sup> boutons either positive (arrow) or negative (arrowhead) for Pclo-Znf1-mCh. Right: Images depicting FM1-43 unloading kinetics in *Pclo*<sup>wt/wt</sup> and *Pclo*<sup>gt/gt</sup> boutons either positive (arrow) or negative (arrowhead) for Pclo-Znf1-mCh. (D) Quantification of (C). The presence of Pclo-Znf1-mCh is sufficient to partially rescue slowed FM1-43 unloading kinetics and total FM1-43 dye released in *Pclo*<sup>gt/gt</sup> boutons (*Pclo*<sup>wt/wt</sup> = 52.64 ± 1.04, n = 213 synapses; *Pclo*<sup>gt/gt</sup> = 40.23 ± 1.55, n = 144 synapses; *Pclo*<sup>wt/wt</sup> + Pclo-Znf1-mCh = 48.7 ± 0.96, n = 257 synapses; *Pclo*<sup>gt/gt</sup> + Pclo-Znf1-mCh = 48.74 ± 0.87, n = 201 synapses; four independent experiments). (E) Model illustrating the contribution of Piccolo in the recycling of synaptic vesicles. Left panel: Piccolo regulates Rab5 function and subsequently early endosome formation through its interaction with Pra1. In the absence of Piccolo (right panel) less Pra1 is localized within the presynaptic terminal. Less synaptic Pra1 negatively impacts Rab5 function, slowing early endosome formation with the consequence that SVs pools and synaptic activity are decreased over time. Scale bars represent 5 µm. Error bars in bar graph represent 95% confidence intervals. Numbers given represent mean ± SEM. ANOVA with Tukey multi comparison test. * denotes $p<0.05$, *** denotes $p<0.001$ and **** denotes $p<0.0001$.

DOI: https://doi.org/10.7554/eLife.46629.030

The following source data is available for figure 11:

**Source data 1.** This spreadsheet contains the normalized values used to generate the bar plots shown in *Figure 11B and D*.
DOI: https://doi.org/10.7554/eLife.46629.031

---

nor leads to an increase in the steady-state levels of endocytic structures (*Mukherjee et al., 2010*). The continued expression of lower molecular weight variants retaining Znf1 and/or Znf2 and thus the synaptic recruitment of Pra1 could explain these differences (see below).

## Synaptic transmission in boutons lacking Piccolo

A fundamental question is how the loss of Piccolo and the reduced number of SVs adversely impacts synapse function. Our electrophysiological data indicate that the evoked release of neurotransmitter is not affected (*Figure 3*). This is somewhat surprising as Piccolo has been shown to physically interact with AZ proteins including RIM, RimBP, VGCC and Munc13 (*Gundelfinger et al., 2015*), which are known to be involved in SV priming and calcium mediated SV fusion (*Breustedt et al., 2010*; *Davydova et al., 2014*; *Girach et al., 2013*; *Schoch et al., 2002*). However, a function in synaptic release could easily be masked by complementary roles played by other AZ proteins such as Bassoon (*Gundelfinger et al., 2015*).

Although RRP, Pvr and PPR were not altered (*Figure 3*), we did observe an enhanced rundown of EPSC amplitudes during 10 Hz stimulation in boutons lacking Piccolo (*Figure 3*). Conceptually, this could be related to a faster depletion of a smaller reserve pool of SVs or defects in the retrieval of SVs from the plasma membrane. Reduced levels of FM4-64 dye after KCl stimulation support the idea that the TRP of SVs is smaller in *Pclo*<sup>gt/gt</sup> neurons (*Figure 3*). Furthermore, the larger number of endocytic-like vesicles (*Figure 2*) suggests that loss of Piccolo impairs efficient recycling of SVs, reducing the pool of SVs. Accordingly, FM1-43 unloading in boutons lacking Piccolo was dramatically slowed indicating that SVs are not efficiently regenerated within this time frame (*Figure 3*).

## Early endosome trafficking is impaired in boutons lacking Piccolo

An important question is how Piccolo loss of function impairs SV regeneration/recycling? Interestingly, the loss/inactivation of proteins regulating SV endocytosis, like Endophilin, Synaptojanin, Dynamin and the small GTPase Rab5, cause similar phenotypes to those observed in *Pclo*<sup>gt/gt</sup> neurons (*Hayashi et al., 2008*; *Milosevic et al., 2011*; *Schuske et al., 2003*; *Wucherpfennig et al., 2003*). However, our data with GFP-2x-FYVE argues against a block in early events of endocytosis as a) GFP-2x-FYVE levels are not significantly altered in *Pclo*<sup>gt/gt</sup> neurons (*Figure 4—figure supplement 2*) and b) PI3P is only generated on already pinched off vesicles (*Harris et al., 2000*; *Mani et al., 2007*; *Verstreken et al., 2003*). Importantly, GFP-2x-FYVE puncta are smaller in *Pclo*<sup>gt/gt</sup> boutons (*Figure 4—figure supplement 2*). This indicates that while the initial uptake of SV membranes is not affected, the subsequent formation of the early endosome compartment is disturbed. Of note, our EM analysis revealed the accumulation of round vesicles with a diameter >60 nm, which could represent early endocytic vesicles (*Figure 2*).

In line with this concept, we observed that loss of Piccolo differentially affected the appearance of markers for early and late endosomes. Rab5 puncta accumulate along *Pclo*<sup>gt/gt</sup> axons, whereas EEA1 and Rab7 puncta numbers are decreased (*Figure 4*), indicating that the loss of Piccolo leads to an early pre EEA1 block in the maturation of the endosome compartment. Consistently, the

synaptic levels of the endosome proteins EEA1 and Rab7 are decreased in *Pclo$^{gt/gt}$* neurons (*Figure 4*), supporting the hypothesis that Piccolo is important for recruitment of endosome proteins towards synapses. As Rab5-positive early endocytic vesicles are not able to mature into EEA1 positive early endosomes, they accumulate at the synapse and eventually leave it without gaining an early endosome identity, indicated by increased numbers of GFP-Rab5-positive puncta along *Pclo$^{gt/gt}$* axons (*Figure 4*). Additionally, lower synaptic levels of mchRab7 as well as reduced numbers of GFP-Rab7 positive puncta along axons indicate that trafficking of proteins through late endosomal structures could be affected. This would subsequently affect the efficient endo-lysosomal degradation of defective molecules.

PI3P as well as Rab5-GTP are necessary to promote the formation of early endosomes (*Spang, 2009*). PI3P levels are not altered in *Pclo$^{gt/gt}$* neurons (*Figure 4—figure supplement 2*), suggesting a possible defect in the activation of Rab5. It is activated, amongst others, via its GEF Rabex5 (*Horiuchi et al., 1997*). Interestingly, levels of Rabex5 at GFP-2x-FYVE puncta are reduced in *Pclo$^{gt/gt}$* neurons (*Figure 5—figure supplement 1*). There are also fewer organelles harboring a complex of Rabex5 and Rab5 (*Figure 5—figure supplement 1*). Importantly, this complex is necessary for the activation of Rab5, the recruitment of EEA1 and the formation of early endosomes (*Murray et al., 2016*; *Simonsen et al., 1998*; *Stenmark et al., 1995*). Alterations in Rabex5 could lead to less active Rab5 and fewer early endosomes, a concept further supported by the fact that the expression of a GDP-locked Rab5 (Rab5$^{S34N}$) in *Pclo$^{wt/wt}$* neurons also causes a reduction in EEA1 at GFP-2x-FYVE puncta and thus fewer early endosomes (*Figure 6*). However Rab5$^{S34N}$ expression does lead to more GFP-2x-FYVE/Rab5 double positive organelles indicating that the inactive GTPase still becomes associated with PI3P-positive organelles. This is different in *Pclo$^{gt/gt}$* neurons, where already less Rab5 is present at PI3P-positive organelles, indicating that the loss of Piccolo may cause less efficient recruitment of Rab5 onto endosome membranes (*Figures 5* and *6*). Consistently, a constitutive active Rab5 (Rab5$^{Q79L}$) rescued EEA1 level at GFP-2x-FYVE puncta (*Figure 7*). It also rescued Synaptophysin levels back to WT levels, indicating that Rab5 dependent early endosome formation contributes to SV pool size at excitatory vertebrate synapses (*Figure 7*). This notion is consistent with studies in *Drosophila*, where Rab5 dominant negative constructs also block the maturation of early endosomes at the neuromuscular junction (NMJ), while the overexpression of WT Rab5 increases SV endocytosis and quantal content (*Wucherpfennig et al., 2003*).

## The formation of early endosomes depends on synaptic activity

Although the molecular mechanisms regulating SV endocytosis have been studied in great detail (*Saheki and De Camilli, 2012*), less is known about the subsequent endocytic steps and their relationship to synaptic transmission and SV recycling. Important open questions include whether the early endosome compartment is part of the SV cycle and thus activity dependent? Hoopmann and colleagues could show that SV proteins travel through an endosome compartment, while they are recycled (*Hoopmann et al., 2010*). Furthermore, it has been shown at the *Drosophila* NMJ that the synaptic FYVE-positive compartment disappears upon block of synaptic activity (*Wucherpfennig et al., 2003*). Consistently, we could show in rat hippocampal neurons that synaptic levels of EEA1 significantly decrease upon TTX treatment (*Figure 8—figure supplement 1*), indicating that early endosome formation in vertebrate synapses is also activity dependent.

## Links between Piccolo and the endosome pathway

Our data show that Rab5 activity and subsequently the recruitment of EEA1 towards PI3P containing membranes, depends on the presence of Piccolo (*Figure 5*). A still open question is how Piccolo can influence Rab5 function and the formation of early endosomes. So far a prominent role for Piccolo for presynaptic F-actin assembly has been described (*Waites et al., 2011*). Yet neither stabilizing F-actin in *Pclo$^{gt/gt}$* neurons rescued the recruitment of Rab5 or EEA1 to GFP-2x-FYVE puncta (*Figure 8—figure supplement 2*) nor destabilizing F-actin assembly in *Pclo$^{wt/wt}$* neurons mimicked endosome phenotypes observed in *Pclo$^{gt/gt}$* neurons (*Figure 8—figure supplement 2*). This suggests that the Piccolo associated endosome phenotype is not due to its role in F-actin assembly, though we cannot rule out that it contributes in some parts to the SV recycling phenotype. Further studies are necessary to elucidate this question.

Interestingly, early interaction studies have shown an interaction between Piccolo and the GDI replacement factor (GDF) Pra1 (*Fenster et al., 2000*), a protein known to promote Rab5 recruitment to endosome membranes where it becomes activated (*Abdul-Ghani et al., 2001*; *Fenster et al., 2000*; *Hutt et al., 2000*; *McLauchlan et al., 1998*). Pra1 nicely localizes at synapses (*Figure 9*) and is thus a good candidate to regulate synaptic Rab5 activity. In line with this concept, synaptic levels of Pra1 are reduced in $Pclo^{gt/gt}$ synapses (*Figure 9*). Importantly, the expression of a single Piccolo Znf1 domain, which binds Pra1 and localizes to presynaptic boutons (*Figure 10*, *Figure 9—figure supplement 1*) (*Fenster et al., 2000*), was sufficient to restore presynaptic levels of both Pra1 and EEA1 (*Figure 10*). These data indicate that Piccolo promotes the localization and stabilization of Pra1 within presynaptic boutons and thus Rab5/EEA1 dependent formation of early endosomes.

## Defects in early endosome formation lead to impaired SV cycling at synapses

Our study of Piccolo loss of function neurons reveals consequences to endosome compartments as well as SV cycling. A remaining question is whether the observed defects in the formation of early endosomes are causal for the observed defects in SV cycling. The expression of inactive mCh-Rab5$^{S34N}$ in $Pclo^{wt/wt}$ neurons also reduced the numbers of early endosomes (*Figure 6*) and mimicked functional defects in SV cycling, seen in $Pclo^{gt/gt}$ neurons. The fact that inactive Rab5 negatively impacts early endosome numbers and SV cycling supports the hypothesis that the formation of early endosomes and SV cycling are functionally linked (*Figure 11*). In this case, one would also expect that rescuing synaptic levels of endosome proteins through the expression of Rab5$^{Q79L}$ and/or Pclo-Znf1-mCh (*Figure 8*) would be sufficient to restore functional SV recycling. In fact, both are able to restore the size of the TRP of SVs in $Pclo^{gt/gt}$ neurons (*Figure 8*). In contrast to Rab5$^{Q79L}$, Pclo-Znf1-mCh is also able to speed up FM1-43 unloading kinetics in $Pclo^{gt/gt}$ boutons (*Figure 11*). Surprisingly, dominant active Rab5$^{Q79L}$ does not have such a positive effect on SV cycling (*Figure 8*). The differences between Rab5$^{Q79L}$ and Pclo-Znf1-mCh in $Pclo^{gt/gt}$ neurons could be due to the fact that Rab5$^{Q79L}$ cannot cycle anymore. It has been shown that active cycling of Rab5 is necessary for vesicle fusion (*Wucherpfennig et al., 2003*). Taken together our functional data support the hypothesis that the dynamic formation of early endosome compartments at synapses is functionally important for efficient SV cycling and that Piccolo, Pra1 and Rab5 are key regulators.

Finally, it is worth considering the disease relevance of the observed changes at $Pclo^{gt/gt}$ synapses. As mentioned above, Piccolo has been implicated in psychiatric, developmental and neurodegenerative disorders, though the molecular mechanism and its contribution to these disorders is unclear. The current study provides insights into one possible underlying mechanism. Specifically, the observed defects in early endosome formation are anticipated to affect the normal trafficking of SV proteins and membranes. This could adversely affect the functionality of synapses throughout the brain, by impairing synaptic transmission, the maintenance of SV pools and the integrity of synapses. Anatomically, these alterations could destabilize synapses, leading for example, to neuronal atrophy during aging, a concept consistent with recent studies showing that Piccolo along with Bassoon regulates synapse integrity through the endo-lysosome and autophagy systems (*Okerlund et al., 2017*; *Waites et al., 2013*). Future studies will help clarify these relationships.

## Materials and methods

**Key resources table**

| Reagent type (species) or resource | Designation | Source or reference | Identifiers | Additional information |
| --- | --- | --- | --- | --- |
| Antibody | anti-synaptophysin (mouse monoclonal) | Synaptic systems | RRID:AB_887824 | (1:1000) |
| Antibody | anti-synaptophysin (guinea pig polyclonal) | Synaptic systems | RRID:AB_1210382 | (1:1000) |
| Antibody | anti-synapsin (rabbit pig polyclonal) | abcam | RRID:AB_1281135 | (1:200) |

*Continued on next page*

*Continued*

| Reagent type (species) or resource | Designation | Source or reference | Identifiers | Additional information |
|---|---|---|---|---|
| Antibody | anti-VGlut1 (guinea pig polyclonal) | Synaptic systems | RRID:AB_887878 | (1:1000) |
| Antibody | anti-Rab5 (mouse monoclonal) | Synaptic systems | RRID:AB_887773 | (1:500) |
| Antibody | anti-Rab7 (rabbit monoclonal) | abcam | RRID:AB_2629474 | (1:500) |
| Antibody | anti-EEA1 (rabbit monoclonal) | Cell signaling | RRID:AB_2096811 | (1:200) |
| Antibody | anti-GFP (chicken polyclonal) | Thermo scientific | RRID:AB_2534023 | (1:500) |
| Antibody | anti-GFP (mouse monoclonal) | Sigma-Aldrich | RRID:AB_390913 | (1:500) |
| Antibody | anti-Rabex5 (rabbit polyclonal) | Thermo scientific | RRID:AB_11157010 | (1:100) |
| Antibody | anti-MAP2 (chicken polyclonal) | Millipore | RRID:AB_571049 | (1:1000) |
| Antibody | anti-Piccolo (rabbit polyclonal) | synaptic systems | RRID:AB_887759 | (1:500) |
| Antibody | anti-Piccolo (rabbit polyclonal) | abcam | RRID:AB_777267 | (1:1000) |
| Antibody | anti-Piccolo (guinea pig polyclonal) | synapitc system | RRID:AB_2619831 | (1:500) |
| Antibody | anti-Bassoon (rabbit polyclonal) | synapitc systems | RRID:AB_88769 | (1:1000 |
| Antibody | anti-Dynamin (rabbit monoclonal) | abcam | RRID:AB_869531 | (1:1000 |
| Antibody | HRP-conjugated secondary | Thermo scientific | | (1:1000 |
| Antibody | Alexa 488, 555, 642 -conjugated secondaries | Thermo scientific | | (1:1000) |
| Recombinant DNA reagent | pLenti-CMV-Neo-Dest | Adgene | plasmid 17392 | used for subcloning |
| Recombinant DNA reagent | pCDH-EF1a-MCS-(PGK-PURO) | System Bioscience | CD810A-1 | used for subcloning |
| Recombinant DNA reagent | pLenti-CMV-Puro-Dest | abcam | plasmid 17452 | used for subcloning |
| Recombinant DNA reagent | pLenti-CMV-GFP-2x-FYVE-Ne | this paper | | |
| Recombinant DNA reagent | pLenti_CMV_mcherry-Rab5_S34N_Puro | this paper | | |
| Recombinant DNA reagent | pLenti_CMV_mcherry-Rab5_Q79L_Puro | this paper | | |
| Recombinant DNA reagent | FU Znf1_Pclo_mcherry | *Waites et al., 2013* | DOI: 10.1038/emboj.2013.27 | |
| Recombinant DNA reagent | FU GFP-Rab5 | this paper | | |
| Recombinant DNA reagent | FU GFP-Rab7 | this paper | | |

*Continued on next page*

*Continued*

| Reagent type (species) or resource | Designation | Source or reference | Identifiers | Additional information |
|---|---|---|---|---|
| Recombinant DNA reagent | pCDH_EF1a_GFP-2xFYVE_IRES_mcherryRab5_Q79L | this paper | | |
| Recombinant DNA reagent | pCDH_EF1a_GFP-2xFYVE_IRES_mcherry-Rab5_S34N | this paper | | |
| Sequence-based reagent | F2-Pclo sequencing primer | this paper | | 5'gcaggaacacaaaccaacaa3' |
| Sequence-based reagent | R1-Pclo sequencing primer | this paper | | 5' tgacctttagccggaactgt3' |
| Sequence-based reagent | SBF2-Pclo sequencing primer | this paper | | 5'tcatcaaggaaaccctggac3' |
| Chemical compound, drug | Tetrodotoxin | abcam | ab120054 | |
| Chemical compound, drug | FM4-64 dye | Thermo scientific | T3166 | |
| Chemical compound, drug | FM1-43 dye | Thermo scientific | T3163 | |
| Chemical compound, drug | Latrunculin | abcam | ab144290 | |
| Chemical compound, drug | Jasplakinolide | Sigma-Aldrich | Sigma-Aldrich | |
| Genetic reagent () | Piccolo KO rat (Pclo rat) | *Medrano et al., 2018* | DOI: https://doi.org/10.1101/405985 | |
| Strain, strain background () | wistar rat | | RRID:RGD_13508588 | femal and male Pclo rats were used for this study |

## Hippocampal cell culture preparation

Micro-island cultures were prepared from hippocampal neurons and maintained as previously described (*Arancillo et al., 2013*). All procedures for experiments involving animals, were approved by the animal welfare committee of Charité Medical University and the Berlin state government. Hippocampi were harvested from $Pclo^{wt/wt}$ and $Pclo^{gt/gt}$ (Wistar) P0-2 rats of either sex. Neurons were plated at 3000 cells/35 mm well on mouse astrocyte micro-islands to generate autaptic neurons for electrophysiology experiments.

For live cell imaging and immunocytochemistry, hippocampal neuron cultures were prepared using the Banker culture protocol (*Banker and Goslin, 1988*; *Meberg and Miller, 2003*; *Tanaka, 2002*). Astrocytes derived from mouse P0-2 cortices were plated into culture dishes 5–7 d before adding neurons. Nitric acid treated coverslips with paraffin dots were placed in separate culture dishes and covered with complete Neurobasal-A containing B-27 (Invitrogen, Thermo Fisher scientific, Waltham, USA), 50 U/ml Penicillin, 50 μg/ml streptomycin and 1x GlutaMAX (Invitrogen, Thermo Fisher scientific, Waltham, USA). $Pclo^{wt/wt}$ and $Pclo^{gt/gt}$ hippocampi were harvested from P0-2 brains in ice cold HBSS (Gibco, Thermo Fisher scientific, Waltham, USA) and incubated consecutive in 20 U/ml papain (Worthington, Lakewood, USA) for 45–60 min and 5 min in DMEM consisting of albumin (Sigma-Aldrich, St. Louis, USA), trypsin inhibitor (Sigma-Aldrich, St.Louis, USA) and 5% FCS (Invitrogen, Thermo Fisher scientific, Waltham, USA) at 37°C. Subsequent tissue was transferred to complete Neurobasal-A, and triturated. Isolated cells were plated on coverslips with paraffin dots at a density of 100,000 cells/35 mm well and 50,000 cells/20 mm well. After 1,5 hr coverslips were flipped upside down and transferred to culture plates containing astrocytes in complete Neurobasal-A. When necessary, neurons were transduced with lentiviral constructs 72–94 hr after plating. Cultures were incubated at 37°C, 5% $CO_2$ for 10–14 days before starting experiments.

## Genotyping Pclo rats

Ear pieces taken from rats were digested over night at 50°C in SNET-buffer (400 mM NaCl, 1% SDS, 200 mM Tris (pH 8.0), 5 mM EDTA) containing 10 mg/ml proteinase K. Subsequently samples were incubated for 10 min at 99°C and centrifuged for 2 min at 13.000 rpm. DNA was precipitated from the supernatant with 100% isopropanol, centrifuged for 15 min at 13.000 rpm, washed with 70% ethanol and centrifuged again for 10 min at 13.000 rpm. The obtained pellet was re-suspended in $H_2O$. PCR reaction with a specific primer combination was performed to determine genotypes. The following primers were used: F2: 5'gcaggaacacaaaccaacaa3'; R1: 5' tgacctttagccggaactgt3'; SBF2: 5'tcatcaaggaaaccctggac3'. The PCR reaction protocol was the following: 2 min 94°C; 3 x (30 s 94 °C, 60 °C 30 s, 72 °C 30 s); 35 x (94 °C 30 s, 55 °C 30 s, 72 °C 30 s); 72 °C 10 min.

## Plasmid constructions

Rab5 and Rab7 were obtained from Richard Reimer (Stanford University) and subcloned inframe with the C-terminus of GFP as BsrG1-EcoR1 fragments in the lentiviral vector FUGWm (*Leal-Ortiz et al., 2008*). The Znf1-Pclo-mCh construct was used from an earlier study (*Waites et al., 2013*). Third generation lentiviral vectors were created using plasmids previously described (*Campeau et al., 2009*). GFP-2x-FYVE (*Gillooly et al., 2000*) was cloned into an ENTRY vector and transferred into pLenti-CMV-Neo-Dest (gift from Eric Campeau, Addgene plasmid # 17392) by a gateway LR recombination. For co-expression of GFP-2x-FYVE with mCh-Rab5 (Q79L), we generated a bi-cistronic expression cassette linking GFP-2x-FYVE and mCh-Rab5 (Q79L) by an IRES sequence. The resulting ENTRY vectors were transferred by Gateway LR recombination into a gateway-enabled destination vector derived from pCDH-EF1a-MCS-IRES-PURO (SystemBiosciences), which lacked the IRES sequence and the puromycin resistance cassette. For co-expression of GFP-2x-FYVE with mCh-Rab5 (S34N), we generated a bi-cistronic expression cassette linking GFP-2x-FYVE and mCh-Rab5 (S34N) by an IRES sequence. The resulting ENTRY vectors were transferred by Gateway LR recombination into a gateway-enabled destination vector derived from pCDH-EF1a-MCS-IRES-PURO (SystemBiosciences), which lacked the IRES sequence and the puromycin resistance cassette. mCh-Rab5 (Q79L) and mCh-Rab5 (S34N) were PCR-generated and subcloned into ENTRY vectors; lentiviral expression vectors were generated by Gateway LR recombination into pLenti-CMV-Puro-Dest (gift from Eric Campeau, Addgene plasmid # 17452).

## Lentivirus production

Lentivirus was produced as previously described (*Haferlach and Schoch, 2002*). In brief, an 80% confluent 75 cm$^2$ flask of HEK293T cells was transfected with 10 µg shuttle vector and mixed helper plasmids (pCMVd8.9 7.5 µg and pVSV-G 5 µg) using XtremeGene 9 DNA transfection reagent (Roche Diagnostics, Mannheim, Germany). After 48 hr, cell culture supernatant was harvested and cell debris was removed by filtration. Aliquots of the filtrate were flash frozen in liquid nitrogen and stored at −80°C until use. Viral titer was estimated by counting cells in mass culture WT hippocampal neurons expressing GFP or mCherry (mCh) as fluorescent reporter. Primary hippocampal cultures were infected with 80 µl of the viral solution ($0.5-1 \times 10^6$ IU/ml) 72–96 hr post-plating.

## Western blot analysis

Brains from P0–P2 animals as well as hippocampal neurons 14 days in vitro (DIV) were lysed in Lysis Buffer (50 mM Tris-HCl, 150 mM NaCl, 5 mM EDTA, 1% TritonX-100, 0.5% Deoxycholate, protease inhibitor pH 7.5) and incubated on ice for 5 min. Samples were centrifuged at 4°C with 13.000 rpm for 10 min, supernatant was transferred into a fresh tube and the protein concentration was determined using a BCA protein assay kit (Thermo Fisher scientific, Waltham, Massachusetts, USA). The same protein amounts were separated by SDS-Page and transferred onto nitrocellulose membranes (running buffer: 25 mM Tris, 190 mM glycin, 0.1% SDS, pH 8.3; transfer buffer: 25 mM Tris, 192 mM Glycine, 1% SDS, 10% Methanol for small proteins, 7% Methanol for larger proteins pH 8.3). Afterwards membranes were blocked with 5% milk in TBST (20 mM Tris pH 7.5, 150 mM NaCl, 0.1% Tween 20) and incubated with primary antibodies in 3% milk in TBST o.n. at 4°C. The following antibodies were used: Piccolo (1:1000; rabbit; synaptic systems, Göttingen, Germany; Cat# 142002, RRID:AB_887759), Piccolo (1:1000; rabbit; abcam, Cambridge, UK; Cat# ab20664, RRID:AB_777267), Piccolo (1:1000, guinea pig, synaptic systems, Göttingen, Germany; Cat# 142104, RRID:

AB_2619831), Synaptophysin (1:1000; mouse; synaptic systems, Göttingen, Germany; Cat# 101011, RRID:AB_887824), VGlut1 (1:1000; guinea pig; synaptic systems, Göttingen, Germany; Cat# 135304, RRID:AB_887878), Rab5 (1:1000; mouse; synaptic systems, Göttingen, Germany; Cat# 108011, RRID: AB_887773), Rab7 (1:1000; rabbit; abcam, Cambridge, UK; Cat# ab137029, RRID:AB_2629474), EEA1 (1:1000; rabbit; cell signaling, Danvers, USA; Cat# 3288S, RRID:AB_2096811), Bassoon (1:1000; rabbit; synaptic systems, Göttingen, Germany; Cat# 141002, RRID:AB_887698), Dynamin (1:1000; rabbit; abcam, Cambridge, UK; Cat# ab52611, RRID:AB_869531). The following day membranes were washed three times with TBST and incubated with HRP labeled secondary antibodies for 1 hr at RT (Thermo Fisher scientific, Waltham, USA, dilution 1:1000). Subsequently membranes were washed three times with TBST and secondary antibody binding was detected with ECL Western Blotting Detection Reagents (Thermo Fisher scientific, Waltham, Massachusetts, USA) and a Fusion FX7 image and analytics system (Vilber Lourmat).

## Electrophysiology

Whole-cell voltage-clamp experiments were performed using one channel of a MultiClamp 700B amplifier (Molecular Devices, Sunnyvale, USA) under control of a Digidata 1440A Digitizer (Molecular Devices, Sunnyvale, USA) and pCLAMP Software (Molecular Devices, Sunnyvale, USA). Neurons were recorded at DIV 12–18. Neurons were clamped at −70 mV, EPSCs were evoked by 2 ms depolarization to 0 mV, resulting in an unclamped AP. Data were sampled at 10 kHz and Bessel filtered at 3 kHz. Series resistance was typically under 10 MΩ. Series resistance was compensated by at least 70%. Extracellular solution for all experiments, unless otherwise indicated, contained (in mM): 140 NaCl, 2.4 KCl, 10 HEPES, 10 glucose, 4 $MgCl_2$, 2 $CaCl_2$ (pH 7.4). Intracellular solution contained the following (in mM): 126 KCl, 17.8 HEPES, 1 EGTA, 0.6 $MgCl_2$, 4 MgATP, 0.3 $Na_2GTP$, 12 creatine phosphate, and phosphocreatine kinase (50 U/mL) (300 mOsm, pH7.4). All reagents were purchased from Carl Roth GMBH (Essen, Germany) with the exception of $Na_2ATP$, sodium $Na_2GTP$, and creatine-phosphokinase (Sigma-Aldrich, St. Louis, MO), and Phosphocreatine (EMD Millipore Chemicals, Billerica, USA).

Electrophysiological recordings were analyzed using Axograph X (Axograph, Berkley, USA), Excel (Microsoft, Redmond, USA), and Prism software (GraphPad, La Jolla, USA). EPSC amplitude was determined as the average of 5 EPSCs at 0.1 Hz. RRP size was calculated by measuring the charge transfer of the transient synaptic current induced by a 5 s application of hypertonic solution (500 mM sucrose in extracellular solution). Pvr was determined as a ratio of the charge from evoked EPSC and the RRP size of the same neuron. Short-term plasticity was analyzed evoking 50 synaptic responses at 10 Hz. PPR was measured dividing the second EPSC amplitude with the first EPSC amplitude elicited with an inter-pulse-interval of 25 ms.

## Electron microscopy

For electron microscopy, neurons were grown on 6 mm sapphire disks and processed as earlier described (*Watanabe et al., 2013*). In brief, cells were sandwiched between spacers and fixed through high-pressure freezing. Following freeze substitution with 1% osmium and 1% uranylacetate in acetone, sapphire discs were washed four times with acetone and postfixed with 0.1% uranylacetate for 1 hr. Following four washing steps with acetone, cells were infiltrated with plastic (Epon, Sigma-Aldrich). Cells were incubated consecutively with 30% Epon in acetone for 1–2 hr, 70% Epon in acetone for 2 hr and 90% Epon in acetone o.n at 4°C. The next day, cells were further infiltrated with pure Epon for 8 hr. Epon was changed three times. Later the plastic was polymerized at 60°C for 48 hr. 70 nm sections were cut using a microtome (Reichert Ultracut S, Reichert/Leica, Wetzlar, Germany) and collected on 0.5% formvar grids (G2200C, Plano, Wetzlar Germany). The sections were stained with 2.5% uranyl acetate in 70% methanol for 5 min prior to imaging. Sections were imaged on a TM-Zeiss-900 electron microscope equipped with a 1 k slow scan CCD camera (TW 7888, PROSCAN, Germany). Presynaptic structures were identified by morphology. Image analysis was performed using custom macros (ImageJ). Briefly, vesicle and endosome structures were outlined with the 'freehand' selection tool and area and length was measured. In addition a line was drawn along the PSD and measured. Docked vesicles were counted manually. Vesicles were defined as docked when the vesicle membrane touched the plasma membrane. Statistical significance was determined using Student's t-test with GraphPad Prism.

## Immunocytochemistry (ICC)

Hippocampal neurons DIV14-16 were prepared for ICC. In brief, cells growing on coverslips were washed in PBS and subsequently fixed in 4% paraformaldehyde (PFA) for 5 min at RT. After washing in phosphate buffered saline (PBS, Thermo Fisher scientific, Waltham, USA), cells were permeabilized with 0.1% Tween20 in PBS (PBS-T). Afterwards cells were blocked in blocking solution (5% normal goat serum in PBS-T) for 30 min and incubated with primary antibodies in blocking solution overnight at 4°C. The following antibodies were used: Synaptophysin (1:1000; mouse; synaptic systems, Göttingen, Germany; Cat# 101011, RRID:AB_887824), Synaptophysin (1:1000; guinea pig; synaptic systems, Göttingen, Germany; Cat# 101004, RRID:AB_1210382), Synapsin (1:200; rabbit; abcam, Cambridge, UK; Cat# ab64581, RRID:AB_1281135), VGlut1 (1:1000; guinea pig; synaptic systems, Göttingen, Germany; Cat# 135304, RRID:AB_887878), Rab5 (1:500; mouse; synaptic systems, Göttingen, Germany; Cat# 108011, RRID:AB_887773), Rab7 (1:500; rabbit; abcam, Cambridge, UK; Cat# ab137029, RRID:AB_2629474), EEA1 (1:200; rabbit; cell signaling, Danvers, USA; Cat# 3288S, RRID:AB_2096811), GFP (1:500; chicken; Thermo Scientific, Waltham, USA; Cat# A10262, RRID:AB_2534023), GFP (1:500; mouse; Roche, Basel, Switzerland; Cat# 11814460001, RRID:AB_390913), Rabex5 (1:100; rabbit; Thermo scientific, Waltham, USA; Cat# PA5-21117, RRID:AB_11157010) MAP2 (1:1000; chicken; Millipore, Darmstadt, Germany; Cat# AB5543, RRID:AB_571049), Piccolo (1:500; rabbit; synaptic systems, Göttingen, Germany; Cat# 142002, RRID:AB_887759), Piccolo (1:1000; rabbit; abcam, Cambridge, UK; Cat# ab20664, RRID:AB_777267), Piccolo (1:500, guinea pig, synaptic systems, Göttingen, Germany; Cat# 142104, RRID:AB_2619831). Afterwards cells were washed three times with PBS-T and incubated with secondary antibodies for 1 hr at RT. Differently labeled secondary antibodies were used from Invitrogen (Thermo Fisher Scientific, Waltham, USA, dilution 1:1000). After washing, coverslips were mounted in ProLong Diamond Antifade Mountant (Thermo Fisher scientific, Waltham, USA).

## Rat perfusion

Adult rats of different ages were first bemused in Isoflurane (Abbott GmbH and Co. KG, Wiesbaden, Germany) and then deeply anesthetized with a mix of 20 mg/ml Xylavet (CO-pharma, Burgdorf, Germany) and 100 mg/ml Ketamin (Inresa Arzneimittel GmbH, Freiburg, Germany) in 0.9% NaCl (B/BRAUN, Melsungen, Germany). Afterwards the heart was made accessible by opening the thoracic cavity. Subsequently a needle was inserted into the protrusion of the left ventricle and the atrium was cut open with a small scissor. Thereby the blood could be exchanged with PBS. Following the exchange of blood, the animals were perfused with freshly made 4% PFA. Following fixation, the brain was dissected from the head and further incubated in 4% PFA for 24 hr at 4°C. Subsequently the brain was transferred to 15% sucrose at 4°C for cryo-protection. After the tissue sank down to the bottom of the tube, it was transferred into 30% sucrose and incubated again until sinking to the bottom. Afterwards the brains were frozen using 2-methylbutane (Carl-Roth, Karlsruhe, Germany) cooled with dry ice to −60°C and stored at −20°C until cut with a cryostat.

## Brain sectioning

20 μm thin sections were cut from frozen brains using a Leica cryostat. Brains were attached to a holder in sagittal as well as coronal orientation and cut at − 20°C. Single sections were transferred from the blade of the knife onto a cooled slide. Slides were stored at −20°C.

## Immunohistochemistry (IHC)

20 μm thin $Pclo^{wt/wt}$ or $Pclo^{gt/gt}$ brain sections were taken out of the freezer and dried for 2 hr at RT. Subsequently sections were surrounded with a liquid barrier marker (AN92.1, Carl-Roth, Karlsruhe, Germany) to prevent solutions from overflowing and washed 3 times for 10 min in TBST (20 mM Tris-base, 150 mM NaCl, 0.025% Triton X-100). Subsequently, sections were blocked for 2 hr in TBS plus 10% normal goat serum and 1% BSA and incubated with primary antibodies in TBS plus 1% BSA overnight in a humidity chamber at 4°C. The next day, sections were washed 3 times for 5 min with TBST before adding the secondary fluorophore-conjugated antibody in TBST 1% BSA for 1 hr at RT. After three washing steps in TBS, sections were counterstained for 30 min (1:1000 DAPI in TBS) and washed 2 times for 10 min in TBS. Afterwards, sections were mounted in ProLong Diamond Antifade Mountant (Thermo Fisher scientific, Waltham, USA).

## Super-resolution imaging using structured illumination

SIM imaging was performed on a Deltavision OMX V4 microscope equipped with three water-cooled PCO edge sCMOS cameras, 405 nm, 488 nm, 568 nm and 642 nm laser lines and a × 60 10.42 numerical aperture Plan Apochromat lens (Olympus). Z-Stacks covering the whole cell, with sections spaced 0.125 µm apart, were recorded. For each Z-section, 15 raw images (three rotations with five phases each) were acquired. Final super-resolution images were reconstructed using softWoRx software and processed in ImageJ/FIJI.

## FM-dye uptake

Functional presynaptic terminals were labeled with FM4-64 dye (Invitrogen, Thermo Fisher scientific, Waltham, USA) as described earlier (*Waites et al., 2013*). In brief, $Pclo^{wt/wt}$, $Pclo^{wt/wt}$ expressing mChRab5$^{S34N}$ or mChRab5$^{Q79L}$ or Pclo-Znf1-mCh as well as $Pclo^{gt/gt}$, $Pclo^{gt/gt}$ expressing mChRab5$^{S34N}$ or Rab5$^{Q79L}$ or Pclo-Znf1-mCh neurons (12–16 DIV) were mounted in a custom-built imaging chamber and perfused with Tyrode's saline solution (25 mM HEPES, 119 mM NaCl, 2.5 mM KCl, 30 mM glucose, 2 mM CaCl$_2$, 2 mM MgCl$_2$, pH 7.4) at 37°C. An image of the basal background fluorescence was taken before the addition of the FM dye. To define the total recycling pool of SV, neurons were incubated with Tyrode's buffer containing 1 µg/ml FM4-64 or FM1-43 dye and 60 or 90 mM KCl for 90 or 60 s. Subsequently, unbound dye was washed off and images were acquired from different areas of the coverslip.

## FM-dye unloading

DIV 21 $Pclo^{wt/wt}$ or $Pclo^{wt/wt}$ expressing mChRab5$^{S34N}$ or Rab5$^{Q79L}$ or Pclo-Znf1-mCh as well as $Pclo^{gt/gt}$ or $Pclo^{gt/gt}$ expressing mChRab5$^{S34N}$ or Rab5$^{Q79L}$ or Pclo-Znf1-mCh primary hippocampal neurons were mounted in a field stimulation chamber (Warner Instruments, Hamden, USA) and perfused with Tyrode's saline solution (25 mM HEPES, 119 mM NaCl, 2.5 mM KCl, 30 mM glucose, 2 mM CaCl2, 2 mM MgCl2, pH 7.4) at 37°C. At first, an image was taken to obtain background fluorescence. Afterwards, Tyrode's saline solution was exchanged with Tyrode's saline solution plus 1 µg/ml FM1-43 dye and neurons were stimulated with 900 AP 10 Hz using a field stimulator (Warner Instruments, Hamden, USA), which was controlled by pCLAMP Software (Molecular Devices, Sunnyvale, USA). Following stimulation, unbound dye was washed away with at least 10 ml Tyrode's saline solution passing through the chamber. Following washing, five images were taken to document FM1-43 dye uptake after stimulation. Afterwards cells were stimulated again with 900 AP and 10 Hz. 120 images were taken every second to document unloading of the FM1-43 dye during stimulation.

## Image acquisition

Images following immunocytochemical staining as well as during FM uptake experiments were acquired on a spinning disc confocal microscope (Zeiss Axio Observer.Z1 with Andor spinning disc unit and cobolt, omricron, i-beam lasers (405, 490, 562 and 642 nm wavelength)) using a 63 × 1.4 NA Plan-Apochromat oil objective and an iXon ultra (Andor, Belfast, UK) camera controlled by iQ software (Andor, Belfast, UK). For live cell imaging, neurons (DIV14–16) growing on 22 × 22 mm coverslips were mounted in a custom-built chamber designed for perfusion, heated to 37°C by forced-air blower and perfused with Tyrode's saline solution (25 mM HEPES, 119 mM NaCl, 2.5 mM KCl, 30 mM glucose, 2 mM CaCl$_2$, 2 mM MgCl$_2$, pH 7.4).

Images from FM unloading experiments were acquired on a Olympus IX83 microscope (Olympus, Hamburg, Germany) equipped with a 60 × 1.2 NA UPlanSApo water objective, a CoolLED system (405 nm, 470 nm, 555 nm, 640 nm wavelength) (Acal$^{bfi}$, Göbenzell, Germany) and a Andor Zyla camera (Andor, Belfast, UK).

## Image processing

For image processing ImageJ/FIJI (*Schindelin et al., 2012*) and OpenView software (written by Dr. Noam Ziv, Technion Institute, Haifa, Israel) was used. Piccolo intensity in VGlut1 puncta as well as FM4-64 dye uptake in $Pclo^{wt/wt}$ and $Pclo^{gt/gt}$ neurons was measured using a box routine with OpenView software. To measure Synaptophysin, Synaptotagmin, VGlut1 and Synapsin intensities in $Pclo^{wt/wt}$ and $Pclo^{gt/gt}$ neurons, 9-pixel regions of interest (ROIs) positive for Synaptophysin were manually selected in ImageJ, subsequently the mean intensity within these ROIs was measured using

a customized ImageJ script. To determine the number of synapses, the number of Synapsin puncta along dendrites was calculated from randomly picked MAP2 positive primary dendrites. Puncta per unit length of dendrite were counted manually. The number of GFP-Rab5, GFP-Rab7 and EEA1 positive puncta along axons was determined from randomly picked axon sections. Numbers of puncta per unit length of axon were counted manually. The intensity of various endosome proteins (Rabex5, Rab5, Rab7, EEA1, Pra1) at synapses was measured using a customized ImageJ script. 9-pixel ROIs were manually picked based on Synaptophysin staining. Subsequently, corresponding fluorescence intensities were measured in all active channels. The background fluorescence was subtracted from all ROIs before the average intensity of the endosome proteins was calculated from all selected ROIs. The fraction of synapses positive for endosome proteins was calculated with a defined intensity value as threshold. It was defined as the standard derivation intensity of the corresponding endosome protein measured in $Pclo^{wt/wt}$ synapses. The size of GFP-2x-FYVE positive compartments imaged with SIM was measured using ImageJ. The 'freehand' tool was used to mark and measure the area of GFP-2x-FYVE puncta. The intensity of endosome proteins (Rabex5, Rab5, EEA1) at GFP-2x-FYVE puncta was measured using a customized ImageJ script. 9-pixel ROIs were picked manually based on GFP-2x-FYVE staining along axons. Subsequently, corresponding fluorescence intensities were measured in all active channels. The background fluorescence was subtracted from all ROIs before the average intensity of the endosome protein was calculated from all selected eGFP-2x-FYVE puncta. The fraction of double or triple positive GFP-2x-FYVE vesicles was calculated using a defined intensity value as threshold. This was defined as the standard derivation intensity of the corresponding protein intensity measured in $Pclo^{wt/wt}$ synapses. Synaptophysin intensity in the presence or absence of Rab5$^{Q79L}$ or Rab5$^{S34N}$ was determined in ImageJ. 9-pixel ROIs were picked along axons positive or negative for GFP-2x-FYVE. Subsequently, Synaptophysin intensity within the defined ROIs was measured using a customized ImageJ script. The intensity of different endosome proteins at the cell soma was measured using ImageJ. The 'free-hand' tool was used to surround the soma. Subsequently the fluorescence intensity within this area was measured and depicted as intensity per soma area. The intensity of different endosome proteins along dendrites was measured using ImageJ. 9-pixel ROIs were picked manually along dendrites marked by MAP2. Subsequently, corresponding fluorescence intensities were measured using a customized ImageJ script.

FM1-43 uptake intensities in mCh-Rab5$^{S34N}$, mCh-Rab5$^{Q79L}$ or Pclo-Znf1-mCh positive or negative synapses were measured in FM1-43 single or FM1-43 – mCh-Rab5$^{S34N}$/mCh-Rab5$^{Q79L}$/Pclo-Znf1-mCh double positive puncta from the same coverslip in ImageJ. 9-pixel ROIs were picked, subsequently FM1-43 intensities within defined ROIs were measured using a customized ImageJ script.

FM1-43 unloading kinetics in mCh-Rab5$^{S34N}$, mCh-Rab5$^{Q79L}$ or Pclo-Znf1-mCh positive or negative synapses were measured in FM1-43 single or FM1-43 – mCh-Rab5$^{S34N}$/mCh-Rab5$^{Q79L}$/Pclo-Znf1-mCh double positive puncta from the same coverslip in ImageJ. ROIs were picked manually and subsequently FM1-43 intensities within the defined ROIs were measured throughout the time series using a customized ImageJ script. The percentage of FM1-43 dye released during 90 s field stimulation was calculated analyzing the remaining FM1-43 dye intensity after 92 s. The difference between the beginning of the stimulation and the end was than calculated as % released FM1-43 dye.

Post-processing of automatically measured image data used 'python' and the 'pandas' data analysis package (*McKinney, 2010*). Statistical analysis was calculated in GraphPad Prism. Data points were plotted using python, matplotlib and seaborn (*Hunter, 2007*).

## Statistical analysis

All values are shown as means ± 95% confidence interval. Statistical significance was assessed using Student's *t* test or ANOVA for comparing multiple samples. For all statistical tests, the 0.05 confidence level was considered statistically significant. In all figures, * denotes $p < 0.05$, ** denotes $p < 0.01$, *** denotes $p < 0.001$ and **** denotes $p < 0.0001$.

## Acknowledgements

We would like to thank Prof. Eckart D Gundelfinger for discussion and valuable comments on the manuscript, Anny Kretschmer for technical assistance and the Virus Core facility, Charité Berlin. We

also thank the Advanced Light Microscopy core facility at the Radium Hospital in Oslo for access to an OMX super-resolution microscope.

## Additional information

### Funding

| Funder | Grant reference number | Author |
|---|---|---|
| Deutsche Forschungsge-meinschaft | SFB958 | Craig Curtis Garner |
| European Research Council | ERC Advanced | Christian Rosenmund |
| Norwegian Cancer Society | Early carrier development grant | Kay Oliver Schink |
| National Institutes of Health | R01HD053889 | F Kent Hamra |
| National Center for Research Resources | R24RR03232601 | F Kent Hamra |
| NIH Office of the Director | R24OD011108 | F Kent Hamra |
| European Research Council | ERC Advanced ERC-2011-AdG 294742 | Zsuzsanna Izsvák |
| Bundesministerium für Bildung und Forschung | NGFN-2 NGFNplus -ENGINE | Zsuzsanna Izsvák |
| National Institutes of Health | R01HD061575 | F Kent Hamra |

The funders had no role in study design, data collection and interpretation, or the decision to submit the work for publication.

### Author contributions

Frauke Ackermann, Conceptualization, Data curation, Formal analysis, Validation, Investigation, Writing—original draft, Writing—review and editing; Kay Oliver Schink, Software, Investigation, Writing—review and editing; Christine Bruns, Formal analysis, Investigation; Zsuzsanna Izsvák, F Kent Hamra, Resources; Christian Rosenmund, Supervision, Funding acquisition; Craig Curtis Garner, Conceptualization, Supervision, Funding acquisition, Writing—original draft, Project administration, Writing—review and editing

### Author ORCIDs

Frauke Ackermann https://orcid.org/0000-0003-3037-8672
Kay Oliver Schink http://orcid.org/0000-0002-5903-4059
Christian Rosenmund http://orcid.org/0000-0002-3905-2444

### Ethics

Animal experimentation: All procedures for experiments involving animals, were approved by the animal welfare committee of Charité Medical University and the Berlin state government (protocol number: T0036/14, O0208/16).

### Decision letter and Author response

Decision letter https://doi.org/10.7554/eLife.46629.034
Author response https://doi.org/10.7554/eLife.46629.035

## Additional files

### Supplementary files

• Transparent reporting form
DOI: https://doi.org/10.7554/eLife.46629.032

## Data availability

All data generated or analysed during this study are included in the manuscript.

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
