## [Decision Letter]

[Editors’ note: a previous version of this study was rejected after peer review, but the authors submitted for reconsideration. The first decision letter after peer review is shown below.]

Thank you for choosing to send your work, "Critical role for Piccolo in Synaptic Vesicle Retrieval", for consideration at *eLife*. Your submission has been assessed by two reviewers and the evaluation has been overseen by a Reviewing Editor and a Senior Editor. Although the work is of interest, we regret to inform you that the findings at this stage are too preliminary for further consideration at *eLife*.

As you will note from the reviewers’ comments there is lack of enthusiasm for many different reasons. If you are willing to address the majority of these concerns, we may reconsider a new submission. However, note that a detailed rebuttal letter addressing all the issues raised will first be evaluated by the Reviewing Editor. If satisfactory, the paper will be sent out for review as a new submission.

*Reviewer #1:*

The authors describe a characterization of a new rodent piccolo mutant – this time in rats versus the previously published work in mice. The initial physiology, EM and FM1-43 experiments support a modest defect in synaptic vesicle recycling in the mutant, with little effect on baseline synaptic transmission. These figures are compelling and indicate an interesting potential defect. The rest of the paper analyzes early endosomal compartment proteins to suggest a role for Piccolo in the formation of early endosomes through a Par-1 mediated regulation of Rab5 biology. This part of the paper suffers from the fact that most of the experimental findings reflect disturbances of these compartments along axons, not synapses. In addition, there is no physiology, EM or dye uptake experiments to support any of the manipulations done in the last half of the paper, reducing my enthusiasm for this part of the story, along with my ability to interpret the findings.

Essential revisions:

1) I found the physiology data in Figure 3E the most compelling aspect of the phenotype that supports a role in endocytosis. The FM1-43 release data in Figure 3K and the EM data in Figure 2C,F provides support for this interpretation. However, the remaining figures are far less convincing – they show usually quite modest changes in antibody staining or GFP expression for several endosomal proteins, mostly in axons. Many of these figures are quite hard for me to interpret. Therefore, the authors really need to use the physiology, EM and FM1-43 experiments to support the other arguments they make throughout the paper. In particular, they need to do the physiology and FM1-43 experiments on the manipulations they make with Rab5 (Rab5QL) to mimic the phenotype, and especially for the Rab5^Q79L^ and ZnF Piccolo rescue experiments – which should show rescue of the physiology and FM1-43 defects. I appreciate the EM is much more involved, so I don't consider that to be an essential addition (though it would be nice). It seems imperative to link the potential endosome biogenesis problems with functional synaptic readouts for the latter manipulations.

2) In line with my concerns above, why would a change in Rab5, Rab7 and EEA1 along axons directly reflect SV recycling defects at synapses? This connection is unclear and I'm not sure how it can be directly linked to SV trafficking defects versus other putative causes (altered neuronal activity and synaptic growth factor trafficking, altered synapse development, reduced synaptic calcium influx, etc.). This becomes more problematic to interpret when considering the data in Figure 4F, where the levels of synaptic Rab5 is unchanged, though it is altered along axons. Why was Rab7 not tested at synapses – again this is far more relevant to SV cycling than the reduction in axons. Is Piccolo also present in the axon at these sites? Given the proposed link between the binding of Piccolo to Pra1 via its ZnF domain, unless that defective interaction is also occurring along axons, it’s unclear to me how to interpret the axonal staining that form the bulk of the supporting data for the 2nd half of the manuscript. Indeed, where does the ZnF domain end up when overexpressed – at synapses, along axons, or everywhere?

3) Quantification of protein levels is needed in Piccolo mutants for the Westerns shown in Figure 1F. The authors indicate decreased expression and increased expression of several proteins, but without quantification one cannot make that determination on a single western blot. Indeed, it looks like EEA1 levels are more reduced in the heterozygous mutant than the homozygote. This figure is difficult to interpret without quantification from multiple westerns.

4) There is no quantification for the staining done in Figure 1G, where the authors indicate Piccolo is not required for synapse formation. This cannot be concluded from showing staining where there are 1000s of synapses in culture. If there was a modest reduction in synapse number, it would be missed without quantification.

5) The authors use anti-synaptophysin staining in Figure 3F to indicate reduced synaptic vesicle density in their mutant cultures. However, this seems problematic based on their Figure 1F western blots, where although synaptophysin intensity is reduced, another SV protein VGlut1 is not. Assuming those findings hold up to proper quantification of multiple westerns, something doesn't add up here. Perhaps Synaptophysin is reduced through another mechanism rather than through a loss of SVs, given VGlut is normal. The authors need to address this and include other synaptic vesicle markers such as Synaptotagmin 1 and Synaptobrevin 2 in their analysis for both Figure 1F and Figure 3F/G.

*Reviewer #2:*

Ackermann and colleagues analyzed a Piccolo KO rat that appeared to remove a large fraction of the high and low molecular weight isoforms of Piccolo. This is a more complete knockout than the mouse Piccolo KO, in which only the high molecular weight isoforms are removed. Using this rat, the authors discovered a role for Piccolo in the formation or maturation of early endosomes and the recycling of synaptic vesicle. This is a very interesting new function for an active zone protein.

Functionally, the authors showed that Piccolo KO neurons exhibited faster depression of neurotransmitter release and slower destaining of FM dye than WT neurons during a high-frequency train of stimulation. Mechanistically, the authors went on to show that the interaction between Piccolo Znf1 domain and Pra1(a GDI replacement factor) may be critical for the function of Piccolo in the formation or maturation of early endosomes. However, I find that the direct and causal evidence for this conclusion is not very strong.

First, there is no direct evidence to show that the physiological defects of Piccolo KO neurons (i.e., neurotransmitter release and FM dye destaining) are caused by an impairment of the endosome pathway. One experiment that can potentially strengthen the conclusion and is within the expertise of the authors is to examine if Znf1 can also rescue the physiological defects of Piccolo KO neurons (i.e., neurotransmitter release and FM dye destaining).

Second, even though Znf1 binds to Pra1 and can rescue the reduction of Syph, Pra1, and EEA1 in Piccolo KO neurons, but it is possible that Znf1 works through a different mechanism. Ideally, one needs to generate a Znf1 mutant that does not bind to Pra1 anymore, and then examine if this Znf1 mutant fails to rescue. If the authors do not have such a mutant already, then this experiment may be outside of the scope of this study.

[Editors’ note: what now follows is the decision letter after the authors submitted for further consideration.]

Thank you for submitting your article "Critical role for piccolo in synaptic vesicle retrieval" for consideration by *eLife*. Your article has been reviewed by Catherine Dulac as the Senior Editor, a Reviewing Editor, and two reviewers. The reviewers have opted to remain anonymous.

The reviewers have discussed the reviews with one another and the Reviewing Editor has drafted this decision to help you prepare a revised submission.

*Reviewer #1:*

In this new manuscript, Ackermann and colleagues provided additional evidence showing that the expression of Znf1 can rescue the physiological defects of Piccolo KO neurons. Even though the direct evidence linking the effect of Znf1 to Pra1 is still missing, the current manuscript supports a new role for Piccolo in the formation/maturation of early endosomes and the recycling of synaptic vesicle.

The authors are encouraged to carefully proofread and edit the text and figures. For example, in the new Figure 1, panel F (this is from the old supplementary Figure 1), the DAPI is no longer shown, yet the merged images contain the DAPI channel and the legend doesn't explain this either. For Figure 1G, one has to assume that the grey bars represent the heterozygous mutants. It would be much better to clearly indicate the groups in the figures or legends.

*Reviewer #2:*

The authors have added supporting data that extends the prior work. However, the new data makes the manuscript difficult to follow at times due to the dense writing. Some polishing and shortening of the text would be helpful for the readers, as it currently is very dense and hard to get through.

---

## [Author Response]

[Editors’ note: the author responses to the first round of peer review follow.]

As you will note from the reviewers’ comments there is lack of enthusiasm for many different reasons. If you are willing to address the majority of these concerns, we may reconsider a new submission. However, note that a detailed rebuttal letter addressing all the issues raised will first be evaluated by the Reviewing Editor. If satisfactory, the paper will be sent out for review as a new submission.Reviewer #1:[…] 1) I found the physiology data in Figure 3E the most compelling aspect of the phenotype that supports a role in endocytosis. The FM1-43 release data in Figure 3K and the EM data in Figure 2C,F provides support for this interpretation. However, the remaining figures are far less convincing – they show usually quite modest changes in antibody staining or GFP expression for several endosomal proteins, mostly in axons. Many of these figures are quite hard for me to interpret. Therefore, the authors really need to use the physiology, EM and FM1-43 experiments to support the other arguments they make throughout the paper. In particular, they need to do the physiology and FM1-43 experiments on the manipulations they make with Rab5 (Rab5QL) to mimic the phenotype, and especially for the Rab5^Q79L^ and ZnF Piccolo rescue experiments – which should show rescue of the physiology and FM1-43 defects. I appreciate the EM is much more involved, so I don't consider that to be an essential addition (though it would be nice). It seems imperative to link the potential endosome biogenesis problems with functional synaptic readouts for the latter manipulations.

This is a great suggestion. We agree that our study provided rather indirect evidence for the link between the observed endosome phenotype and physiological impairments in Piccolo KO neurons. We thus performed additional experiments. First, we expressed a dominant inactive Rab5^S43N^ in WT neurons and analyzed whether an inactive Rab5 also leads to reduced FM1-43 dye uptake and decreases in the FM1-43 unloading kinetics. Indeed, we observe a reduced FM1-43 uptake as well as FM1-43 unloading kinetics (Figure 6, subsection “Generation and characterization of the Piccolo Knockout rat”). As the effect of Rab5^S43N^ expression mimics the physiological phenotypes observed in Piccolo KO neurons, we interpret these data to indicate that impairing the endosome compartment can be causal for the physiological defects seen in Piccolo KO synapses. This concept is further supported by the fact that the expression of Pclo-Znf1-mCh also rescues the physiological defects in Piccolo KO neurons, specifically we find that FM1-43 dye uptake and FM1-43 unloading kinetics are returned to WT levels (Figure 12, subsection “Expression of Pclo-Znf1 rescues synaptic Pra1 and EEA1 levels, synaptic vesicle pool size as well as SV cycling”). Importantly, we find that the expression of dominant active Rab5^Q79L^ also rescues FM1-43 uptake efficiencies in Piccolo KO neurons (Figure 8, subsection “GTPase deficient Rab5 (Rab5^Q79L^) expression rescues EEA1 and Synaptophysin levels back to *Pclowt/wt* amounts”). Taken together, we believe that these newly added physiological experiments strongly support our core hypothesis that the observed impairment in the formation of early endosomes in Piccolo KO neurons is causally related to the observed physiological defects.

2) In line with my concerns above, why would a change in Rab5, Rab7 and EEA1 along axons directly reflect SV recycling defects at synapses?

We understand this concern and agree that changes in Rab5, Rab7 and EEA1 along axons alone are not direct evidence for defects in SV recycling at synapses. However, we believe that the largest fraction of GFP-2x-FYVE, Rab5 and EEA1 organelles analyzed in our studies represent those localized at synapses. To confirm this hypothesis, we quantified the percentage co-localization between GFP-2x-FYVE and Rab5 puncta at synapses marked by Synaptophysin. Here we observed that 63% of the GFP-2x-FYVE puncta (Figure 4—figure supplement 2) and more than 80% of the Rab5 puncta co-localize with Synaptophysin (Figure 4—figure supplement 2) puncta and thus are at synapses. These data indicate that most of our analysis, along axons, indeed represents synaptic endosomes.

This connection is unclear and I'm not sure how it can be directly linked to SV trafficking defects versus other putative causes (altered neuronal activity and synaptic growth factor trafficking, altered synapse development, reduced synaptic calcium influx, etc.).

We agree that there might be other putative causes, which might affect the formation of endosome compartments in Piccolo KO neurons. However, as Piccolo is strictly a pre-synaptic active zone protein and Rab5^Q79L^ is able to rescue the observed defects (Figure 8, subsection “Pools of recycling SVs are reduced but synaptic release properties are not altered in boutons lacking Piccolo”), while Rab5^S43N^ phenocopies Piccolo loss of function (Figure 6, subsection “Generation and characterization of the Piccolo Knockout rat”), we feel the most parsimonious explanation is that Piccolo and Rab5 form a functional link between the presynaptic endosome compartment and recycling SVs.

This becomes more problematic to interpret when considering the data in Figure 4F, where the levels of synaptic Rab5 is unchanged, though it is altered along axons. Why was Rab7 not tested at synapses – again this is far more relevant to SV cycling than the reduction in axons.

This is an excellent suggestion and we agree it would be interesting to examine synaptic Rab7 levels at synapses, as they should be directly linked to altered Rab5 function and impairments in the formation of early endosomes. We thus analyzed the levels of mCh-Rab7 at synapses in Piccolo WT and KO neurons. Here, we observe a significant reduction in synaptic Rab7 levels in neurons lacking Piccolo. This data further supports the hypothesis that the loss of Piccolo leads to defects in synaptic endosome compartments (Figure 4, subsection “Levels of endosome proteins are reduced at *Pclogt/gt* synapses”).

Is Piccolo also present in the axon at these sites? Given the proposed link between the binding of Piccolo to Pra1 via its ZnF domain, unless that defective interaction is also occurring along axons, it’s unclear to me how to interpret the axonal staining that form the bulk of the supporting data for the 2nd half of the manuscript. Indeed, where does the ZnF domain end up when overexpressed – at synapses, along axons, or everywhere?

Our working hypothesis is that the interaction between Piccolo and Pra1 primarily takes place within presynaptic boutons as Piccolo is almost exclusively a presynaptic active zone protein (Cases-Langhoff et al..,1996; Aczonin; Garner Gundelfiner 2016). With regard to the distribution of Pclo-Znf1-mCh, we analyzed the percentage of overlay between Pclo-Znf1-mCh and synapses marked by Synaptophysin (Figure 8—figure supplement 1, subsection “Expression of Pclo-Znf1 rescues synaptic Pra1 and EEA1 levels, synaptic vesicle pool size as well as SV cycling”). Here, we used a low titer of lentivirus (<30% infected neurons) allowing a cell-autonomous distribution of Pclo-Znf1-mCh to be assessed in the axons of infected cells, growing along the dendrites of un-infected neurons. Here we observed that more than 80% of the Pclo-Znf1-mCh puncta co-localize with Synaptophysin, indicating that the Znf1 domain is rather selectively localized to presynaptic boutons. This distribution suggests a primary effect of Pclo-Znf1-mCh on membrane recycling through presynaptic endosomes.

3) Quantification of protein levels is needed in Piccolo mutants for the Westerns shown in Figure 1F. The authors indicate decreased expression and increased expression of several proteins, but without quantification one cannot make that determination on a single western blot. Indeed. It looks like EEA1 levels are more reduced in the heterozygous mutant than the homozygote. This figure is difficult to interpret without quantification from multiple westerns.

This is a great suggestion. We thus quantified several western blots and added it to figure 1 (subsection “Generation and characterization of the Piccolo Knockout rat”). EEA1 levels are most severely affected by the loss of Piccolo. For other synaptic proteins, we only observe minor changes.

4) There is no quantification for the staining done in Figure 1G, where the authors indicate Piccolo is not required for synapse formation. This cannot be concluded from showing staining where there are 1000s of synapses in culture. If there was a modest reduction in synapse number, it would be missed without quantification.

We agree that without quantification we might miss a modes reduction in synapse numbers. We therefore performed a proper analysis, quantifying the number of Synapsin puncta per length of primary dendrites. We could detect no significant difference in the number of Synapsin puncta formed between Piccolo WT and Piccolo KO neurons (Figure 1, subsection “Generation and characterization of the Piccolo Knockout rat”). This is in line with earlier knockdown and knockout studies (Leal-Ortiz et al., 2008; Mukherjee et al., 2010).

5) The authors use anti-synaptophysin staining in Figure 3F to indicate reduces synaptic vesicle density in their mutant cultures. However, this seems problematic based on their Figure 1F western blots, where although synaptophysin intensity is reduced, another SV protein VGlut1 is not. Assuming those findings hold up to proper quantification of multiple westerns, something doesn't add up here. Perhaps Synaptophysin is reduced through another mechanism rather than through a loss of SVs, given VGlut is normal. The authors need to address this and include other synaptic vesicle markers such as Synaptotagmin 1 and Synaptobrevin 2 in their analysis for both Figure 1F and Figure 3F/G.

This is a great suggestion. We thus quantified the levels of additional synaptic vesicle proteins. Similar to Synaptophysin, we observed that both Synaptotagmin and VGlut1 were reduced. However, the levels of Synapsin, which is a peripherally associated SV protein, were not changed (Figure 1—figure supplement 1, subsection “Generation and characterization of the Piccolo Knockout rat”). These new data, together with our EM studies, further support our conclusion that synaptic vesicles pools are reduced in presynaptic terminals from Piccolo KO neurons.

Reviewer #2:Ackermann and colleagues analyzed a Piccolo KO rat that appeared to remove a large fraction of the high and low molecular weight isoforms of Piccolo. This is a more complete knockout than the mouse Piccolo KO, in which only the high molecular weight isoforms are removed. Using this rat, the authors discovered a role for Piccolo in the formation or maturation of early endosomes and the recycling of synaptic vesicle. This is a very interesting new function for an active zone protein.Functionally, the authors showed that Piccolo KO neurons exhibited faster depression of neurotransmitter release and slower destaining of FM dye than WT neurons during a high-frequency train of stimulation. Mechanistically, the authors went on to show that the interaction between Piccolo Znf1 domain and Pra1(a GDI replacement factor) may be critical for the function of Piccolo in the formation or maturation of early endosomes. However, I find that the direct and causal evidence for this conclusion is not very strong.First, there is no direct evidence to show that the physiological defects of Piccolo KO neurons (i.e., neurotransmitter release and FM dye destaining) are caused by an impairment of the endosome pathway. One experiment that can potentially strengthen the conclusion and is within the expertise of the authors is to examine if Znf1 can also rescue the physiological defects of Piccolo KO neurons (i.e., neurotransmitter release and FM dye destaining).

We agree that only indirect evidence is provided by our data that an impairment of the endosome pathway is causal for the observed physiological defects in Piccolo KO neuron. Therefore, we tested as suggested whether expression of Pclo-Znf1 can rescue physiological phenotypes in Piccolo KO neurons.

We observe that the presence of Pclo-Znf1-mCh at synapses is sufficient to rescue FM1-43 dye uptake in Piccolo KO neurons (Figure 12, subsection “Expression of Pclo-Znf1 rescues synaptic Pra1 and EEA1 levels, synaptic vesicle pool size as well as SV cycling”). Furthermore, the presence of Pclo-Znf1 was found to increase FM 1-43 dye unloading kinetics as well as the total amount of FM dye released during a 90sec field stimulation (Figure 12, subsection “Expression of Pclo-Znf1 rescues synaptic Pra1 and EEA1 levels, synaptic vesicle pool size as well as SV cycling”). We believe that these new data provide stronger evidence that impairments in the endosome pathway are causal for the physiological defects observed in Piccolo KO neurons. Please note that we also added new data showing that the expression of dominant inactive Rab5^S43N^ also leads to the physiological defects seen in Piccolo KO neurons (Figure 6, subsection “GDP-locked Rab5 (Rab5S34N) expression in *Pclowt/wt* neurons decreases early endosomes and SV cycling to similar levels seen in *Pclogt/gt* neurons”) further supporting the link between Piccolo, the endosome pathway and SV cycling.

Second, even though Znf1 binds to Pra1 and can rescue the reduction of Syph, Pra1, and EEA1 in Piccolo KO neurons, but it is possible that Znf1 works through a different mechanism. Ideally, one need to generate a Znf1 mutant that does not bind to Pra1 anymore, and then examine if this Znf1 mutant fails to rescue. If the authors do not have such a mutant already, then this experiment may be outside of the scope of this study.

Unfortunately, we have yet to create such a mutant, and though we agree it would be nice to add, the time it would take to generate such reagents place the experiments outside the scope of the present study.

[Editors' note: the author responses to the re-review follow.]

Reviewer #1:In this new manuscript, Ackermann and colleagues provided additional evidence showing that the expression of Znf1 can rescue the physiological defects of Piccolo KO neurons. Even though the direct evidence linking the effect of Znf1 to Pra1 is still missing, the current manuscript supports a new role for Piccolo in the formation/maturation of early endosomes and the recycling of synaptic vesicle.The authors are encouraged to carefully proofread and edit the text and figures. For example, in the new Figure 1, panel F (this is from the old supplementary Figure 1), the DAPI is no longer shown, yet the merged images contain the DAPI channel and the legend doesn't explain this either. For Fig1G, one has to assume that the grey bars represent the heterozygous mutants. It would be much better to clearly indicate the groups in the figures or legends.

We agree with the reviewer’s comment. We carefully proofread the manuscript and the figures. We shortened the text and transferred some of the data into the supplement. Figure 9 representing the effect of activity on synaptic levels of endosome proteins (Rab5, EEA1, Rabex5) is now Figure 8—figure supplement 1. Additionally, we moved data about Rabex5 in the supplements and created a new Figure 5—figure supplement 1. We also removed the DAPI channel in Figure 1F and added a legend in Figure 1G.

Reviewer #2:The authors have added supporting data that extends the prior work. However, the new data makes the manuscript difficult to follow at times due to the dense writing. Some polishing and shortening of the text would be helpful for the readers, as it currently is very dense and hard to get through.

We agree with the reviewer’s comment. We shortened the text and transferred some of the data into the supplement to make the manuscript more concise. Figure 9 representing the effect of activity on synaptic levels of endosome proteins (Rab5, EEA1, Rabex5) is now Figure 8—figure supplement 1. Additionally, we moved data about Rabex5 in the supplements and created a new Figure 5—figure supplement 1.